# HOW MANY SAMPLES ARE NEEDED TO TRAIN A RELU FEED-FORWARD NEURAL NETWORK?

## ABSTRACT

Neural networks have become standard tools in many areas, yet many important statistical questions remain open. This paper studies the question of how much data are needed to train a ReLU feed-forward neural network. Our theoretical and empirical results suggest that the generalization error of ReLU feed-forward neural networks scales at the rate $1/\sqrt{n}$ in the sample size $n$ rather than the usual "parametric rate" $1/n$. Thus, broadly speaking, our results underpin the common belief that neural networks need "many" training samples.

## 1 INTRODUCTION

Neural networks have ubiquitous applications in science and business (Goodfellow et al., 2016; Graves et al., 2013; LeCun et al., 2015; Badrinarayanan et al., 2017). However, our understanding of their statistical properties remains incomplete. An important open question is the number of samples required for training a neural network. More specifically: Can we improve the generalization error rate of a feed-forward neural network from $1/\sqrt{n}$ to $1/n$?

Over the past two decades, significant progress has been made in our theoretical understanding of various aspects of deep neural networks. This progress includes a multitude of research papers focusing on analyzing and deriving upper and lower bounds for the generalization error (L. Bartlett et al., 2017; Arora et al., 2018; Kawaguchi et al., 2017; Neyshabur et al., 2018). The results in Neyshabur et al. (2015); L. Bartlett et al. (2017); Arora et al. (2018); Neyshabur et al. (2017; 2018); Nagarajan & Kolter (2019), highlight the relationship between the complexity of the model (for example, the depth and width of the network) and the generalization error; however, a common limitation is that the generalization bounds tend to exhibit a strong dependence, often exponential, on either the depth of the network or the number of nodes per layer. Golowich et al. (2018) can do away with this direct reliance on the network's depth by assuming norm constraints on the parameter matrix of each layer. Taheri et al. (2021) and Mohades & Lederer (2023), establish an upper bound on the generalization error that exhibits a logarithmic growth in the total number of parameters and the potential for decrease with more layers. Quite interestingly, our lower bound in this paper matches their upper bound. Although these studies collectively contribute to our understanding of the generalization error of deep neural networks, they do not develop matching lower bounds.

In parallel with the aforementioned fields of research, there is a body of research focused on investigating the mini-max lower bounds for deep-Rectified Linear Unit (ReLU) networks (Suzuki, 2018; Imaizumi & Fukumizu, 2019; Parhi & D. Nowak, 2022; Raskutti et al., 2009; Schmidt-Hieber & Bos, 2022; Raskutti et al., 2012; Schmidt-Hieber, 2020; Zhang & Wang, 2023; Tsuji, 2021), but their perspective differs from ours.

Our perspective in this paper, views neural networks as fundamental functions of interest and explores their statistical properties. In contract, they emphasize the distinction between function classes and estimation methods, often comparing neural networks to alternatives like wavelet transforms and kernel methods. The core of their research—which is very similar to Zhang et al. (2002); Donoho & Johnstone (1998) that exploit wavelet threshold estimators— centers around the utilization of deep neural networks to investigate the mini-max lower bounds for estimating nonparametric regression models characterized by sparse additive structures and specific smoothness properties, such as Lipschitz, Hölder, or Sobolev functions. Imaizumi & Fukumizu (2019) provides a comprehensive review of prior research related to function estimation using deep neural networks. Their mini-max lower bounds for the function classes degrade either with the depth of the network or with

the parameters of smoothness. These developments have provided valuable intuition, but establishing comprehensive lower bounds in the mini-max setting for deep neural networks with non-linear activation functions still remains open.

We establish a lower bound on the mini-max risk for deep-ReLU networks using information theory. Our bound scales as $\sqrt{\log(d)/n}$ (with $n$ as the number of training samples and $d$ as the input dimension) and is independent of the network depth or the number of parameters. We also show empirically that this seems the learned rate in practice.

**Our three main contributions are:**

1. We establish that a mini-max risk lower bound for ReLU feed-forward neural networks does not depend on the depth or width of the network except in logarithmic factor. This bound decreases as $1/\sqrt{n}$ with the number of training samples $n$ (Lemma 1).

2. We demonstrate that the space of shallow-ReLU feed-forward networks can be viewed as a subspace of the deep-ReLU feed-forward networks (Lemma 7).

3. We show empirically that the generalization error rate for ReLU feed-forward neural networks can't be improved beyond $1/\sqrt{n}$-rate (Section 4), that supports our theoretical findings.

**Organisation:** Section 2 provides the problem formulation and establishes a lower bound on the mini-max risk for ReLU feed-forward neural networks (Theorem 1). Section 3 provides some technical results that form our main result's foundation including, an upper bound for the mutual information of the packing set of network space (Lemma 4) and a lower bound for the packing number of shallow-ReLU network space (Lemma 6). Section B contains the proofs for Lemma 4. In Section 4, we shift our focus to empirical findings to support our theories. We conclude our paper in Section 5. More technical results, empirical details, and detailed proofs are deferred to the Appendix.

## 2  PROBLEM FORMULATION AND MAIN RESULT

This section provides an outline of the core elements of our study. We introduce the background before presenting our main result. To start, we consider the following regression model

$$y_i = f^*(\boldsymbol{x}_i) + u_i \qquad \text{for } i \in \{1, \ldots, n\} \tag{1}$$

For an unknown neural network $f^* : \mathbb{R}^d \to \mathbb{R}$ and i·i·d· noises $u_i \sim \mathcal{N}(0, \sigma^2)$ with $\sigma \in (0, \infty)$. We observe $n$ i·i·d· data samples $(\boldsymbol{x}_1, y_1), (\boldsymbol{x}_2, y_2), \ldots, (\boldsymbol{x}_n, y_n) \in \mathbb{R}^d \times \mathbb{R}$ drawn independently from a joint distribution $\mathbb{P}_{\boldsymbol{x},y}$ with a fixed marginal distribution $\mathbb{P}_{\boldsymbol{x}} = \mathcal{N}(\boldsymbol{0}, \boldsymbol{I}_d)$. It is assumed that $u_i$ and $\boldsymbol{x}_i$ are independent and that the networks are of the form

$$\begin{aligned} f_{\boldsymbol{\Theta}} &: \mathbb{R}^d \to \mathbb{R} \\ \boldsymbol{x} &\mapsto f_{\boldsymbol{\Theta}}(\boldsymbol{x}) := W^L \boldsymbol{\phi}^L\big(\ldots W^1 \boldsymbol{\phi}^1(W^0 \boldsymbol{x})\big) \end{aligned} \tag{2}$$

indexed by $\boldsymbol{\Theta} = (W^L, \ldots, W^0)$ summarizing the weight matrices $W^l \in \mathbb{R}^{h_{l+1} \times h_l}$ for $l \in \{0, 1, \ldots, L\}$. The number of hidden layers (the depth of the network) is $L \in \{1, 2, \ldots\}$, and $h_l$ denotes the number of nodes in the $l$-th layer (the width of the $l$-th layer), where $h_0 = d$ and $h_{L+1} = 1$. The function $\boldsymbol{\phi}^l : \mathbb{R}^{h_l} \to \mathbb{R}^{h_l}$ is the ReLU activation function of the $l$-th layer which is defined as

$$x \mapsto \max\{0, x\}.$$

We then consider a sparse parameter space $\mathcal{B}$ with $\ell_1$-type constraints on the parameters of the network. We consider $\ell_1$-type constraints as opposed to $\ell_0$-type constraints, primarily because $\ell_0$-type constraints tend to make the problem hard to optimize and "combinatorial", particularly in high-dimensional settings (Lederer, 2022, Chapter 2). Then, we define a function class

$$\mathcal{F} := \{f_{\boldsymbol{\Theta}} : \boldsymbol{\Theta} \in \mathcal{B}\}.$$

The mini-max risk for the function class $\mathcal{F}$, can be defined as (Wainwright, 2019, Chapter 15)

$$\mathcal{R}_{(n,d)}(\mathcal{F}; \Phi \circ \rho) := \inf_{\widehat{f}} \sup_{f^* \in \mathcal{F}} \mathbb{E}_{(\boldsymbol{x}_i, y_i)_{i=1}^n} \Big[ \Phi\big(\rho(\widehat{f}, f^*)\big) \Big], \tag{3}$$

where $\rho : \mathcal{F} \times \mathcal{F} \to [0,\infty)$ is a semi metric[1] and $\Phi : [0,\infty) \to [0,\infty)$ an increasing function. The expectation is taken with respect to the training data $(\boldsymbol{x}_i, y_i)_{i=1}^n$ and the infimum runs over all possible estimators $\widehat{f}$ (measurable functions) of $f^*$ on the training data $(\boldsymbol{x}_i, y_i)_{i=1}^n$. Hence, $\widehat{f}(\boldsymbol{x}) \equiv \widehat{f}(\boldsymbol{x}, \{(\boldsymbol{x}_i, y_i)\}_{i=1}^n)$, where $\boldsymbol{x}$ is a new data point with the same distribution $\mathbb{P}_{\boldsymbol{x}}$. We use the notation $\mathcal{R}_{(n,d)}(\mathcal{F}; \Phi \circ \rho)$ to emphasize that the mini-max risk depends on the number of training samples $n$, the input dimension $d$ and the function space $\mathcal{F}$.

In this paper, our focus is on the standard setting where $\rho$ represents the $L_2(\mathbb{P}_{\boldsymbol{x}})$-norm, and $\Phi(t) = t^2$. Therefore, $\Phi(\rho(\widehat{f}, f^*))$ is the squared $L_2(\mathbb{P}_{\boldsymbol{x}})$-norm, that is our mini-max risk

$$\inf_{\widehat{f}} \sup_{f^* \in \mathcal{F}} \mathbb{E}_{(\boldsymbol{x}_i, y_i)_{i=1}^n} \left[ \|\widehat{f} - f^*\|_{L_2}^2 \right].$$

We assume that the distribution $\mathbb{P}_{\boldsymbol{x}}$ has a density $h(\boldsymbol{x})$ with respect to the Lebesgue measure $d\boldsymbol{x}$ which, implies that

$$\|\widehat{f} - f^*\|_{L_2} := \left( \int_{\boldsymbol{x} \in \mathcal{X}} \left(\widehat{f}(\boldsymbol{x}) - f^*(\boldsymbol{x})\right)^2 h(\boldsymbol{x}) d\boldsymbol{x} \right)^{1/2}.$$

We now present our mini-max risk lower bound for deep-ReLU neural networks. Considering the regression model defined in Equation (1), where $f^* \in \mathcal{F}$ (a ReLU neural network with $L$ hidden layers and $\ell_1$-bounded weights), then we have:

**Theorem 1 (Mini-max risk lower bound for ReLU feed-forward neural networks)** *Using the $L_2(\mathbb{P}_{\boldsymbol{x}})$-norm as our underlying semi metric $\rho$, and $\boldsymbol{x}_1, \dots, \boldsymbol{x}_n \sim \mathcal{N}(\boldsymbol{0}, \boldsymbol{I}_d)$, then for $d \geq 10$ large enough and any increasing function $\Phi : [0,\infty) \to [0,\infty)$, it holds that*

$$\mathcal{R}_{(n,d)}(\mathcal{F}; \Phi \circ \rho) \geq \frac{1}{2} \Phi \left[ c\sqrt{v_1} \left( \frac{\log(d)}{n} \right)^{1/4} \right], \tag{4}$$

*with $c := \sqrt{(\sigma)/(26\kappa)}$, where $\kappa \in [1, \infty)$ is a constant that controls the size of the function space $\mathcal{F}$, and $\|W^L\|_1 := \sum_{k=1}^{h_{L+1}} \sum_{j=1}^{h_L} |W_{kj}^L| \leq v_1$. For $\Phi(\cdot) = (\cdot)^2$, we specifically obtain*

$$\inf_{\widehat{f}} \sup_{f^* \in \mathcal{F}} \mathbb{E}_{(\boldsymbol{x}_i, y_i)_{i=1}^n} \left[ \|\widehat{f} - f^*\|_{L_2}^2 \right] \geq \frac{c^2}{2} v_1 \sqrt{\frac{\log(d)}{n}}. \tag{5}$$

We consider $\kappa$ as a scaling factor that determines the size of the function space $\mathcal{F}$, and we construct a $2\delta$-packing such that, for all pairs of functions $f, f' \in \mathcal{F}$, it holds that $\rho(f, f') \leq 2\kappa\delta$ for $\delta \in (0, \infty)$ and $\kappa \in [1, \infty)$.

Theorem 1 demonstrates that for all possible $\widehat{f}$, risk scales at least as $v_1 \sqrt{\log(d)/n}$, considering an upper bound for $\mathbb{E}_{(\boldsymbol{x}_i, y_i)_{i=1}^n}[\|\widehat{f} - f^*\|_{L_2}^2]$

$$\mathbb{E}_{(\boldsymbol{x}_i, y_i)_{i=1}^n} \left[ \|\widehat{f} - f^*\|_{L_2}^2 \right] \leq \epsilon^2$$

Then, we can reformulate the result of Theorem 1 and conclude that one requires

$$n \geq \left( \frac{c}{\epsilon} \right)^4 \frac{v_1^2 \log(d)}{4},$$

samples to achieve an error of at most $\epsilon^2$.

A related work is M. Klusowski & R. Barron (2017), but there are two important distinctions. First, we allow for ReLU, which is currently the most used activation function, rather than restricting to the more exotic activation functions, such as Sinusoidal, as proposed in the mentioned paper. Second, we consider deep feed-forward neural networks, whereas M. Klusowski & R. Barron (2017) focuses primarily on shallow feed-forward neural networks.

---

[1]A semi metric satisfies all properties of a metric, except that there may exist pairs $f \neq f'$ for which $\rho(f, f') = 0$.

## 3 TECHNICAL RESULTS

Here, we provide technical results essential in proving our main theorem: (i) We leverage an extension of a classical result from information theory, Fano's inequality (Lemma 3), which includes the concept of packing number. This extension involves deriving an upper bound for the mutual information (Lemma 4) and a lower bound for the $\log$ of the packing number of shallow-ReLU networks (Lemma 6). (ii) In Lemma 7, we demonstrate that, under certain constructions, a deep-ReLU network can generate a shallow-ReLU network. Accordingly, we can conclude that the lower bound for the packing number of shallow-ReLU neural network function space can apply to a deep-ReLU neural network function space.

The following notation will be used throughout the paper. For vector $\boldsymbol{v} \in \mathbb{R}^d$, $\ell_0$-norm is defined by $\|\boldsymbol{v}\|_0 := \#\{i \in \{1, \ldots, d\} : v_i \neq 0\}$, $\ell_1$-norm is defined by $\|\boldsymbol{v}\|_1 := \sum_{i=1}^{d} |v_i|$ and the Euclidean norm is defined by $\|\boldsymbol{v}\|_2 := \sqrt{\sum_{i=1}^{d}(v_i)^2}$. We define $\|W^l\|_1 := \sum_{k=1}^{h_{l+1}} \sum_{j=1}^{h_l} |W_{kj}^l|$, for a matrix $W^l \in \mathbb{R}^{h_{l+1} \times h_l}$ where $l \in \{0, 1, \ldots, L\}$. The cardinality of the $2\delta$-packing of the corresponding neural network function space $\mathcal{F}$ for $\delta \in (0, \infty)$ and with respect to $L_2(\mathbb{P}_{\boldsymbol{x}})$-norm is defined as $\mathcal{M} := \mathcal{M}(2\delta, \mathcal{F}, \|\cdot\|_{L_2})$. We define $[\mathcal{M}] := \{1, \ldots, \mathcal{M}\}$ as the index set. And we define $X^n := (\boldsymbol{x}_1, \ldots, \boldsymbol{x}_n)^\top$ and $Y^n := (y_1, \ldots, y_n)^\top$.

We now proceed to provide the definition of packing and covering number (Vaart & Wellner, 1996), which are of great importance in our study.

**Definition 2 (Covering and packing number)** *Consider a metric space consisting of a set $\mathcal{F}$ and a semi metric $\rho$ as defined in Section 2, then,*

> *A) An $2\delta$-covering of $\mathcal{F}$ in the semi metric $\rho$ is a collection $\{f_{\boldsymbol{\Theta}^1}, \ldots, f_{\boldsymbol{\Theta}^{\mathcal{M}}}\} \subseteq \mathcal{F}$ such that for all $f \in \mathcal{F}$, there exists some $i \in [\mathcal{M}]$ with $\rho(f, f_{\boldsymbol{\Theta}^i}) \leq 2\delta$. The $2\delta$-covering number $N(2\delta, \mathcal{F}, \rho)$, is the cardinality of the smallest $2\delta$-covering.*
>
> *B) An $2\delta$-packing of $\mathcal{F}$ in the semi metric $\rho$ is a collection $\{f_{\boldsymbol{\Theta}^1}, \ldots, f_{\boldsymbol{\Theta}^{\mathcal{M}}}\} \subseteq \mathcal{F}$ such that $\rho(f_{\boldsymbol{\Theta}^j}, f_{\boldsymbol{\Theta}^k}) \geq 2\delta$ for all $j, k \in [\mathcal{M}]$ and $j \neq k$. The $2\delta$-packing number $\mathcal{M}(2\delta, \mathcal{F}, \rho)$, is the cardinality of the largest $2\delta$-packing.*

Now, we provide the technical results. Based on the concept of packing and covering number, assume that $\{\mathbb{P}_{f_{\boldsymbol{\Theta}^1}}, \ldots, \mathbb{P}_{f_{\boldsymbol{\Theta}^{\mathcal{M}}}}\}$ is a family of distributions for the corresponding neural networks $f_{\boldsymbol{\Theta}^1}, \cdots, f_{\boldsymbol{\Theta}^{\mathcal{M}}}$ which satisfy $\rho(f_{\boldsymbol{\Theta}^j}(\boldsymbol{x}), f_{\boldsymbol{\Theta}^k}(\boldsymbol{x})) \geq 2\delta$ for all $j, k \in [\mathcal{M}]$ and $j \neq k$. Then, assume that $J$ is uniformly distributed over the index set $[\mathcal{M}]$ and the conditional distribution of $(Y^n | X^n)$ given $J$ defined by $((Y^n | X^n) \mid J = j) \sim \mathbb{P}_{f_{\boldsymbol{\Theta}^j}}$. Then, Fano's inequality (Wainwright, 2019, Proposition 15.12) can be formalized as:

**Lemma 3 (Fano's Inequality)** *Let $\{f_{\boldsymbol{\Theta}^1}, \ldots, f_{\boldsymbol{\Theta}^{\mathcal{M}}}\} \subseteq \mathcal{F}$ be a $2\delta$-packing set respect to $\rho$. Then, for any increasing function $\Phi : [0, \infty) \to [0, \infty)$, the mini-max risk is lower bounded by*

$$\mathcal{R}_{(n,d)}(\mathcal{F}; \Phi \circ \rho) \geq \Phi(\delta) \left( 1 - \frac{I(J; Y^n | X^n) + \log 2}{\log \mathcal{M}(2\delta, \mathcal{F}, \|\cdot\|_{L_2})} \right).$$

The symbol $I(J; Y^n | X^n)$ represents the mutual information between a random index $J$, which is drawn uniformly from the index set $[\mathcal{M}]$ and the samples $(Y^n | X^n)$ drawn from the prior distribution $\mathbb{P}_{f_{\boldsymbol{\Theta}^j}}$ corresponding to $f_{\boldsymbol{\Theta}^j} := f_{\boldsymbol{\Theta}^J}$. The mutual information, measures how much information can be revealed about the index $J$ of a $2\delta$-packing set by drawing the samples $(Y^n | X^n)$.

To apply Fano's inequality, we need the following three lemmas to find 1. an upper bound for the mutual information of the $2\delta$-packing of ReLU-neural network function space $\mathcal{F}$ (Lemma 4) and 2. a lower bound for the $\log$ of the packing number ($\log \mathcal{M}$) for ReLU-neural networks (the combination of Lemma 6 and Lemma 7). We start with upper bounding $I(J; Y^n | X^n)$ of the $2\delta$-packing of neural network function space $\mathcal{F}$ as follows:

**Lemma 4 (Upper bounding $I(J; Y^n | X^n)$ of the $2\delta$-packing of neural network function space $\mathcal{F}$)** *For all possible pairs of two distinct networks $f_{\boldsymbol{\Theta}^j}, f_{\boldsymbol{\Theta}^k} \in \mathcal{F}$ satisfy $\rho(f_{\boldsymbol{\Theta}^j}(\boldsymbol{x}), f_{\boldsymbol{\Theta}^k}(\boldsymbol{x})) \geq 2\delta$, the*

*mutual information $I(J;Y^n|X^n)$ is upper bounded by*

$$I(J;Y^n|X^n) \leq \frac{2n(\kappa\delta)^2}{\sigma^2},$$

*for a suitable $\kappa \in [1,\infty)$, such that $\rho(f_{\boldsymbol{\Theta}^j}(\boldsymbol{x}), f_{\boldsymbol{\Theta}^k}(\boldsymbol{x})) \leq 2\kappa\delta$.*

In the subsequent lemma, our objective is to establish a lower bound for the packing number of shallow-ReLU network function space. Subsequently, we will extend this result to encompass deep-ReLU networks. Consider a shallow neural network with ReLU activation function, denoted as $f_{(W^1,W^0)}$, where

$$f_{(W^1,W^0)}(\boldsymbol{x}) = W^1 \boldsymbol{\phi}^1(W^0\boldsymbol{x}).$$

Recall that $W^1$ and $W^0$ are the weight matrices. We then define a sparse collection of shallow-ReLU networks as $\mathcal{F}_{\mathcal{B}_{\mathrm{Sh}}}$, characterized by

$$\mathcal{F}_{\mathcal{B}_{\mathrm{Sh}}} := \mathcal{F}_{v_0,v_1} := \left\{ f_{(W^1,W^0)} \,|\, (W^1,W^0) \in \mathcal{B}_{v_0,v_1} \right\},$$

where

$$\mathcal{B}_{\mathrm{Sh}} := \mathcal{B}_{v_0,v_1} := \left\{ \|W^0_{j,\cdot}\|_1 \leq v_0, \quad \|\|W^1\|\|_1 \leq v_1 \right\} \quad \text{for all} \quad j \in \{1,\ldots,h_1\},$$

denotes as the corresponding parameter space, where $v_0 \in [1,\infty)$ and $v_1 \in (0,\infty)$.

**Remark 5 (Assumption: $v_0 = 1$)** *For simplicity in the proof of Lemma 6, we assume $v_0 = 1$.*

This assumption is useful for constructing a subclass of function space $\mathcal{F}_{\mathcal{B}_{\mathrm{Sh}}}$ in the proof of Lemma 6 to establish a lower bound for the packing number of a shallow-ReLU network function space. This assumption basically determines the structure of the inner weight (the weight between the input layer and the hidden layer of a shallow neural network). It also guarantees that the number of input dimensions $d$, matches the width of the constructed subclass of $\mathcal{F}_{\mathcal{B}_{\mathrm{Sh}}}$.

Based on the structure of a shallow-ReLU network and the defined corresponding parameter space, our aim for the next lemma is to derive a lower bound for the $\log$ of the packing number $(\log(\mathcal{M}))$ of a shallow-ReLU neural network function space. The key components of this bound are the $\ell_1$-norm control on the parameters of the two layers and the parameter $\delta$, which determines the minimum distance between all possible pairs of two distinct networks $f_{\boldsymbol{\Theta}^j}, f_{\boldsymbol{\Theta}^k} \in \mathcal{F}$ for $j \neq k \in [\mathcal{M}]$. Taking these factors into account, we can conclude the following lemma:

**Lemma 6 (Lower bounding the packing number of shallow-ReLU feed-forward network function space)**
*For a sparse collection of shallow-ReLU feed-forward network function space $\mathcal{F}_{\mathcal{B}_{\mathrm{Sh}}}$, there exist $\delta \in (0,\infty)$ such that*

$$\log \mathcal{M}\left(2\delta, \mathcal{F}_{\mathcal{B}_{\mathrm{Sh}}}, \|\cdot\|_{L_2}\right) \geq \left(\frac{v_1}{13\delta}\right)^2 \log(d).$$

By quantifying the lower bound of the packing number, it provides valuable insights into the capacity and potential complexity of these networks. For small values of $\delta$, a sufficiently wide network becomes necessary. This observation is particularly interesting as it provides valuable insights into selecting an appropriate width for the network based on the input dimension. The larger the input dimension $d$, the wider the network should be.

We then define a sparse collection of deep-ReLU networks denotes as $\mathcal{F}_{\mathcal{B}_{\mathrm{L}}}$, as follows:

$$\mathcal{F}_{\mathcal{B}_{\mathrm{L}}} := \mathcal{F}_{v_L,\ldots,v_0} := \left\{ f_{(W^L,\ldots,W^0)} \,|\, (W^L,\ldots,W^0) \in \mathcal{B}_{\mathrm{L}} \right\},$$

where $\mathcal{B}_{\mathrm{L}}$, denotes the sparse parameter space for deep-ReLU networks and can be defined by

$$\mathcal{B}_{\mathrm{L}} := \left\{ \sum_{l=0}^{L-1} \|\|W^l\|\|_1 \leq v_{\mathrm{s}}, \quad \|\|W^L\|\|_1 \leq v_1 \right\}.$$

We define $v_{\mathrm{s}} = h_1 v_0 + (L-1)\omega$. It's important to emphasize that we are focusing on deep-ReLU neural networks with equal widths for all hidden layers, and this width is equal to that of

the shallow-ReLU neural networks, denoted as $\omega$. Furthermore, based on Remark 5, it holds that $v_{\mathrm{s}} = h_1 + (L-1)\omega$.

In the next lemma, we demonstrate that the function space of shallow-ReLU networks is a subspace of the function space of deep-ReLU networks and establish a lower bound for the packing number of deep-ReLU networks function space, drawing from the earlier established lower bound for shallow-ReLU network function space.

**Lemma 7 (Generating a shallow-ReLU feed-forward network using a deep-ReLU feed-forward network)**
*For ReLU activation functions and defined the shallow and deep function spaces $\mathcal{F}_{\mathcal{B}_{\mathrm{Sh}}}$ and $\mathcal{F}_{\mathcal{B}_{\mathrm{L}}}$, it holds that $\mathcal{F}_{\mathcal{B}_{\mathrm{Sh}}} \subset \mathcal{F}_{\mathcal{B}_{\mathrm{L}}}$. That implies*

$$\log \mathcal{M}\big(2\delta, \mathcal{F}_{\mathcal{B}_{\mathrm{L}}}, \|\cdot\|_{L_2}\big) \geq \log \mathcal{M}\big(2\delta, \mathcal{F}_{\mathcal{B}_{\mathrm{Sh}}}, \|\cdot\|_{L_2}\big).$$

## 3.1 PROOF OF LEMMA 4

**Proof** The aim of this proof is to establish an upper bound on the mutual information $I(J; Y^n | X^n)$, for the $2\delta$-packing within the neural network function space $\mathcal{F}$. To achieve this, we invoke the result of Lemma 10, which establishes a connection between the mutual information and the Kullback-Leibler divergence (KL divergence). In this paper, we use the notation $D_{\mathrm{KL}}(\mathbb{P}_{f_{\Theta^j}} || \mathbb{P}_{f_{\Theta^k}})$ to denote the KL divergence between two probability distributions $\mathbb{P}_{f_{\Theta^j}}$ and $\mathbb{P}_{f_{\Theta^k}}$. We then apply the result obtained from Lemma 9. Finally, we employ the same re-scaling procedure as demonstrated in Wainwright (2019, Example 15.14) and Wainwright (2019, Example 15.16) to construct a $2\delta$-packing in such a way that, for a suitable constant $\kappa \in [1, \infty)$, we ensure that $\rho(f_{\Theta^j}(\boldsymbol{x}), f_{\Theta^k}(\boldsymbol{x})) \leq 2\kappa\delta$ holds for all pairs $f_{\Theta^j}(\boldsymbol{x})$ and $f_{\Theta^k}(\boldsymbol{x})$ corresponding to $j \neq k \in [\mathcal{M}]$.

We can 1. use the result provided by Lemma 10, 2. use the fact that $\sum_{j,k=1}^{\mathcal{M}} D_{\mathrm{KL}}(\mathbb{P}_{f_{\Theta^j}} || \mathbb{P}_{f_{\Theta^k}}) \leq \binom{\mathcal{M}}{2} \sup_{k,j}(D_{\mathrm{KL}}(\mathbb{P}_{f_{\Theta^j}} || \mathbb{P}_{f_{\Theta^k}}))$ for all $j \neq k \in [\mathcal{M}]$, 3. calculate the permutation, 4. some arithmetic calculation, 5. use the fact that $\mathcal{M} \geq 1$, so $0 \leq (\mathcal{M}-1)/\mathcal{M} < 1$, 6. use the view of Lemma 9, 7. invoke the definition of $\rho$ as $L_2(\mathbb{P}_{\boldsymbol{x}})$- norm, 8. employ the re-scaling procedure and 9. simplify the factor 2 to obtain

$$
\begin{aligned}
I(J; Y^n | X^n) &\leq \frac{n}{\mathcal{M}^2} \sum_{\substack{j,k=1 \\ j\neq k}}^{\mathcal{M}} D_{\mathrm{KL}}\big(\mathbb{P}_{f_{\Theta^j}} || \mathbb{P}_{f_{\Theta^k}}\big) \\
&\leq \frac{n}{\mathcal{M}^2} \binom{\mathcal{M}}{2} \sup_{\substack{j,k\in[\mathcal{M}] \\ j\neq k}} \Big(D_{\mathrm{KL}}\big(\mathbb{P}_{f_{\Theta^j}} || \mathbb{P}_{f_{\Theta^k}}\big)\Big) \\
&= \frac{n}{\mathcal{M}^2} \frac{\mathcal{M}!}{(\mathcal{M}-2)!} \sup_{\substack{j,k\in[\mathcal{M}] \\ j\neq k}} \Big(D_{\mathrm{KL}}\big(\mathbb{P}_{f_{\Theta^j}} || \mathbb{P}_{f_{\Theta^k}}\big)\Big) \\
&= \frac{n(\mathcal{M}-1)}{\mathcal{M}} \sup_{\substack{j,k\in[\mathcal{M}] \\ j\neq k}} \Big(D_{\mathrm{KL}}\big(\mathbb{P}_{f_{\Theta^j}} || \mathbb{P}_{f_{\Theta^k}}\big)\Big) \\
&\leq n \sup_{\substack{j,k\in[\mathcal{M}] \\ j\neq k}} \Big(D_{\mathrm{KL}}\big(\mathbb{P}_{f_{\Theta^j}} || \mathbb{P}_{f_{\Theta^k}}\big)\Big) \\
&= \frac{n}{2\sigma^2} \sup_{\substack{j,k\in[\mathcal{M}] \\ j\neq k}} \left(\int_{\boldsymbol{x}\in\mathcal{X}} \big(f_{\Theta^j}(\boldsymbol{x}) - f_{\Theta^k}(\boldsymbol{x})\big)^2 h(\boldsymbol{x}) d\boldsymbol{x}\right) \\
&= \frac{n}{2\sigma^2} \sup_{\substack{j,k\in[\mathcal{M}] \\ j\neq k}} \Big(\rho\big(f_{\Theta^j}(\boldsymbol{x}), f_{\Theta^k}(\boldsymbol{x})\big)^2\Big) \\
&\leq \frac{n(2\kappa\delta)^2}{2\sigma^2} \\
&= \frac{2n(\kappa\delta)^2}{\sigma^2},
\end{aligned}
$$

as desired. ■

## 4 EMPIRICAL STUDIES

The primary objective of this empirical section is to provide concrete evidence to support our theoretical findings. To achieve this, we investigate whether the generalization error of a deep-ReLU neural network scales more significantly with a $1/n$-rate or a $1/\sqrt{n}$-rate. We use "test error" as an estimate of the "generalization error" of a trained deep-ReLU network. We consider both classification and regression tasks, using the $MNIST$ and $CIFAR - 10$ dataset for classification and the California Housing Prices (CHP) dataset for regression analysis. We consider ReLU feed-forward neural networks trained with Cross-entropy (CE) loss and Mean-squared (MS) error for classification and regression dataset, respectively as loss functions (Appendix C). The implementation of these neural networks was carried out using the neural network (nn) package of PyTorch.

We then conduct our empirical studies in two steps: In the first step, we compute the test error of a trained network. In the second step, we determine the appropriate curve (either $1/\sqrt{n}$ or $1/n$ scales) that best fits the test error values. To address the impact of various hyper-parameters of the ReLU neural networks, including the number of training samples $(n)$, network depth $(L)$, and the width of hidden layers, we consider appropriate curves $(c_1 + \alpha/\sqrt{n})$ and $(c_2 + \beta/n)$ with $\alpha, \beta, c_1, c_2 \in (0, \infty)$. Optimizing these parameters is achieved through the Sequential Least Squares Quadratic Programming (SLSQP) method (Kraft, 1988). The "minimize" function from scipy.optimize is employed for SLSQP implementation.

### 4.1 MNIST

The $MNIST$ dataset contains 60,000 training images ($28 \times 28$ pixels) and 10,000 testing images ($28 \times 28$ pixels). According to the size of the images, we have 784 feature inputs. The batch size for the training samples is equal to 100. And the batch size for testing data samples is 10,000. For this dataset, we have increased the number of training samples by the factor of 100. In each step after training the network, all the test data samples (10,000 test samples) have considered, and the loss values are reported (accordingly, we have 600 loss values (it means 600 steps)).

We explore both shallow and deep-ReLU feed-forward neural networks in our experiments. Specifically, we investigate shallow-ReLU neural networks with 5, 10, and 20 hidden nodes. Additionally, we examine a four-hidden layer ReLU feed-forward neural network with a uniform width of 900. As it has shown in Figure 1, Figure 2, Figure 3 and Figure 4, in comparison with $(c_2 + \beta/n)$, $(c_1 + \alpha/\sqrt{n})$ provides a better fit to model the generalization error behavior of the neural networks.

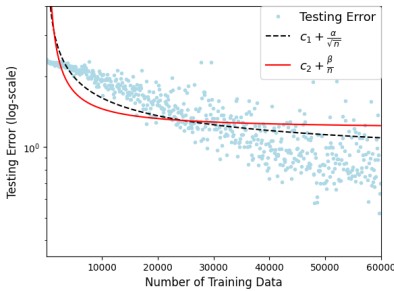

Figure 1: Comparison of the strength of two curves $(c_1 + \alpha/\sqrt{n})$ and $(c_2 + \beta/n)$ to model the generalization error of a shallow-ReLU neural network (with the width of 5) for $MNIST$ dataset

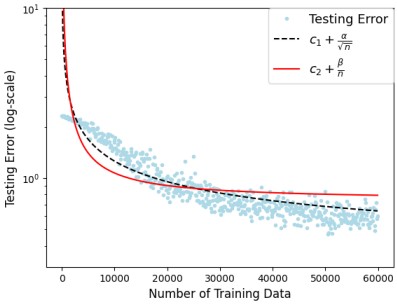

Figure 2: Comparison of the strength of two curves $(c_1 + \alpha/\sqrt{n})$ and $(c_2 + \beta/n)$ to model the generalization error of a shallow-ReLU neural network (with the width of 10) for $MNIST$ dataset

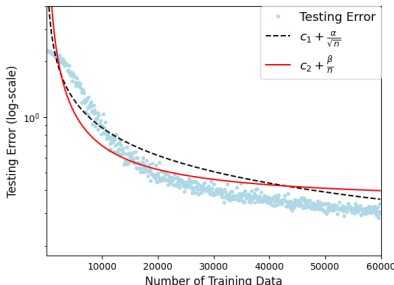

Figure 3: Comparison of the strength of two curves $(c_1 + \alpha/\sqrt{n})$ and $(c_2 + \beta/n)$ to model the generalization error of a shallow-ReLU neural network (with the width of 20) for $MNIST$ dataset

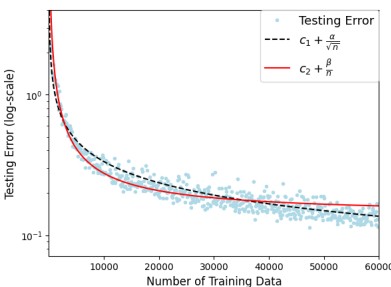

Figure 4: Comparison of the strength of two curves $(c_1 + \alpha/\sqrt{n})$ and $(c_2 + \beta/n)$ to model the generalization error of a deep-ReLU neural network (a four-hidden-layer network with a uniform width of 900) for $MNIST$ dataset

## 4.2 CIFAR-10

The $CIFAR-10$ dataset contains 50,000 training images ($32 \times 32$ color images) and 10,000 testing images ($32 \times 32$ color images). According to the size of the images, we have 3072 feature inputs. The batch size for the training samples is equal to 100. And the batch size for testing data samples is 10,000. For this dataset, we have increased the number of training samples by the factor of 100. In each step after training the network, all the test data samples (10,000 test samples) have considered, and the loss values are reported (accordingly, we have 500 loss values (it means 500 steps)). we examine shallow-ReLU feed-forward neural networks with the widths of 100 and 120. As it has shown in Figure 5, in comparison with $(c_2 + \beta/n)$, $(c_1 + \alpha/\sqrt{n})$ provides a better fit to model the generalization error behavior of the neural networks.

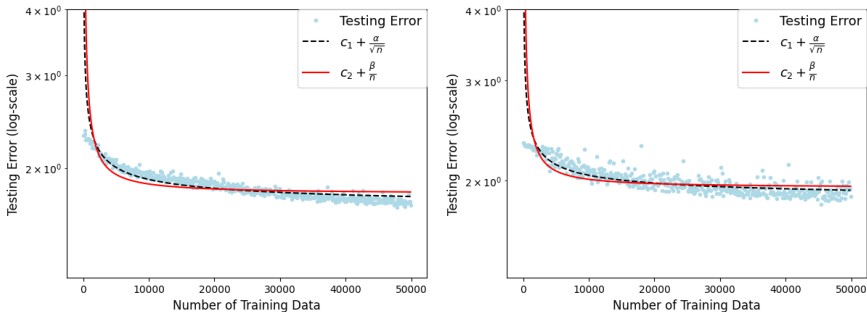

Figure 5: Comparison of the strength of two curves $(c_1 + \alpha/\sqrt{n})$ and $(c_2 + \beta/n)$ to model the generalization error of a shallow-ReLU neural network for $CIFAR - 10$ dataset with the width of 100 and 120 (on the right and left, respectively).

### 4.3 CALIFORNIA HOUSING PRICES (CHP)

The version considered in this study comprises 8 numeric input attributes and a dataset of 20,640 samples. These samples were randomly divided into $75\%$ for training data and the remaining $25\%$ for test data. The batch size for the training samples is set to 20. For this dataset, we increased the number of training samples by a factor of 20. After training the network at each step, all test data samples were considered, and the loss value was recorded. Consequently, we obtained 774 loss values, corresponding to 774 steps. As the testing error stabilized after 120 batches, we compare the results specifically for the first 120 batches. Additionally, the objective function for the SLSQP method also operates on these 120 batches. As it has shown in Figure 6, in comparison with $(c_2 + \beta/n)$, $(c_1 + \alpha/\sqrt{n})$ provides a better fit to model the generalization error behavior of the neural network. We consider a five-hidden layer ReLU neural network with a uniform width of 23.

The values for the parameters of the two curves for both dataset as well as the width of all the hidden layers ($\omega$), the number of hidden layers $L$ and the Learning Rate (LR) are provided in Table 1 (Appendix C).

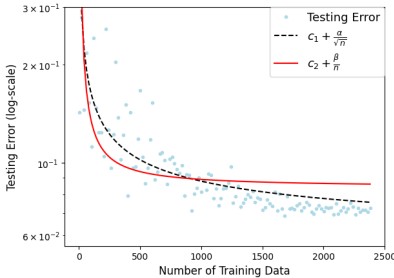

Figure 6: Comparison of the strength of two curves $(c_1 + \alpha/\sqrt{n})$ and $(c_2 + \beta/n)$ to model the generalization error of a deep-ReLU neural network (a five-hidden layer network with a uniform width of 23) for $CHP$ dataset

## 5 CONCLUSION

In this paper, we employ the results from information theory called "Fano's inequality" to establish a mini-max risk lower bound for ReLU feed-forward neural networks that scales at the rate $\sqrt{\log(d)/n}$. This bound indicates that the generalization error of the deep-ReLU feed-forward neural networks cannot be improved beyond a $1/\sqrt{n}$-rate. Our empirical findings support this conclusion and indicate that for both regression and classification problems, the generalization error of ReLU-neural networks scales at the rate $1/\sqrt{n}$.

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

## A    FURTHER TECHNICAL RESULTS

In this section, we present additional technical results from the work of others and our own, that are essential for the proof of Theorem 1's components but might also be of interest by themselves. We divide the results into two main parts. The first part includes a few results from other works that are contained in the proof of Lemma 4, and the second part includes a few results, both from our work and others' to prove Lemma 6.

PART 1: PRELIMINARY RESULTS FOR UPPER BOUNDING THE MUTUAL INFORMATION

We present some auxiliary results that are contained in the proof of Lemma 4. To follow these results more conveniently, we explain the necessary steps briefly. After defining the KL divergence as a measure of distance between two probability measures, we calculate the KL divergence between two multivariate normal distributions (Hayakawa & Suzuki, 2020, Lemma A.1). Then, we calculate KL divergence of $n$-product of two multivariate normal distributions and finally, we find the connection between the mutual information and KL divergence.

The KL divergence (Wainwright, 2019, Equation 3.57) between two different probability distributions $P$ and $Q$ on domain $\mathcal{X}$ with densities $p(\boldsymbol{x})$ and $q(\boldsymbol{x})$ can be defined as

$$D_{\mathrm{KL}}\big(P \,\|\, Q\big) = \int_{\boldsymbol{x}\in\mathcal{X}} p(\boldsymbol{x}) \log \frac{p(\boldsymbol{x})}{q(\boldsymbol{x})} d\boldsymbol{x}\,.$$

As we have $n$ data samples, we are interested to find the KL divergence between two different $n$-product distributions. Assume that $(P^1,\ldots,P^n)$ be a collection of $n$ probability distributions, and define $P^{1:n} := \bigotimes_{i=1}^n P^i$ as the $n$-product distributions. Define another $n$-product distribution $Q^{1:n}$ in a similar way. For the ease of notation, we define $P^n := P^{1:n}$ and $Q^n := Q^{1:n}$. Then, the connection between the KL divergence of $n$-product distributions $P^n$ and $Q^n$ and the KL divergence of the individual pairs (Wainwright, 2019, Equation 15.11a), can be formalized as the following lemma:

**Lemma 8 (Decomposition of the KL divergence for $n$-product distributions)** *For two $n$-product distributions $P^n$ and $Q^n$, it holds that*

$$D_{\mathrm{KL}}\big(P^n \,\|\, Q^n\big) = \sum_{i=1}^n D_{\mathrm{KL}}\big(P^i \,\|\, Q^i\big)\,.$$

*And in the case of i·i·d· product distributions — meaning that $P^i = P^1$ and $Q^i = Q^1$ for all $i \in \{1,\ldots,n\}$— we have*

$$D_{\mathrm{KL}}\big(P^n \,\|\, Q^n\big) = n \times D_{\mathrm{KL}}\big(P^1 \,\|\, Q^1\big)\,.$$

*We consider short-hands $P$ and $Q$, for $P^1$ and $Q^1$, respectively. So, the previous equation takes form*

$$D_{\mathrm{KL}}\big(P^n \,\|\, Q^n\big) = n \times D_{\mathrm{KL}}\big(P \,\|\, Q\big)\,.$$

We then proceed to calculate the KL divergence between two normal distributions. Consider the regression model defined in Equation (1) and the network model defined in Equation (2). We assume that the noise terms are i·i·d· and $u_i \in \mathcal{N}(0,\sigma^2)$. Recall that the explanatory variables $\boldsymbol{x}_i$ follow a fixed distribution $\mathbb{P}_{\boldsymbol{x}}$ and have the density $h(\boldsymbol{x})$. Then, we define $\boldsymbol{z} := (\boldsymbol{x},y) \in \mathbb{R}^d \times \mathbb{R}$ as the joint variable of $\boldsymbol{x}$ and $y$. According to the conditional probability, the joint density can be written as follows:

$$\begin{aligned}
p_{f_{\boldsymbol{\Theta}^j}}(\boldsymbol{z}) &= p_{Y|X}(y|\boldsymbol{x})h(\boldsymbol{x}) \\
&= \frac{1}{\sqrt{2\pi\sigma^2}} e^{-\frac{(y-f_{\boldsymbol{\Theta}^j}(\boldsymbol{x}))^2}{2\sigma^2}} h(\boldsymbol{x})\,,
\end{aligned} \tag{6}$$

where $j \in [\mathcal{M}]$ and $p_{f_{\boldsymbol{\Theta}^j}}(\boldsymbol{z})$ is the joint density of $(\boldsymbol{x},y)$ with regression function $f_{\boldsymbol{\Theta}^j}(\boldsymbol{x})$. And consider $p_{f_{\boldsymbol{\Theta}^k}}(\boldsymbol{z})$ as another joint density of $(\boldsymbol{x},y)$ in the same manner with regression function $f_{\boldsymbol{\Theta}^k}(\boldsymbol{x})$ and two distinct corresponding normal distributions $\mathbb{P}_{f_{\boldsymbol{\Theta}^j}}$ and $\mathbb{P}_{f_{\boldsymbol{\Theta}^k}}$ such that have densities $p_{f_{\boldsymbol{\Theta}^j}}(\boldsymbol{z})$ and $p_{f_{\boldsymbol{\Theta}^k}}(\boldsymbol{z})$, respectively. Recall that $f_{\boldsymbol{\Theta}^j}$ and $f_{\boldsymbol{\Theta}^k}$ are any two distinct neural networks of the neural network model defined in Section 2, which parameterized by $\boldsymbol{\Theta}^j$ and $\boldsymbol{\Theta}^k$ ($j,k \in [\mathcal{M}]$ as any two distinct indices of the $2\delta$-packing set). Then, the KL divergence between any two normal distributions $\mathbb{P}_{f_{\boldsymbol{\Theta}^j}}$ and $\mathbb{P}_{f_{\boldsymbol{\Theta}^k}}$ can be calculated as the following lemma (Yang & Barron, 1999):

**Lemma 9 (The KL divergence between two multivariate normal distributions)** *Assume any two normal distributions $\mathbb{P}_{f_{\boldsymbol{\Theta}^j}}$ and $\mathbb{P}_{f_{\boldsymbol{\Theta}^k}}$ for all $j,k \in [\mathcal{M}]$ and $j \neq k$, then it holds that*

$$D_{\mathrm{KL}}(\mathbb{P}_{f_{\boldsymbol{\Theta}^j}} \,\|\, \mathbb{P}_{f_{\boldsymbol{\Theta}^k}}) = \frac{1}{2\sigma^2}\int_{\boldsymbol{x}\in\mathcal{X}} \big(f_{\boldsymbol{\Theta}^j}(\boldsymbol{x}) - f_{\boldsymbol{\Theta}^k}(\boldsymbol{x})\big)^2 h(\boldsymbol{x})d\boldsymbol{x}$$

*And that*

$$D_{\mathrm{KL}}(\mathbb{P}^n_{f_{\boldsymbol{\Theta}^j}} \| \mathbb{P}^n_{f_{\boldsymbol{\Theta}^k}}) = \frac{n}{2\sigma^2} \int_{\boldsymbol{x} \in \mathcal{X}} \left(f_{\boldsymbol{\Theta}^j}(\boldsymbol{x}) - f_{\boldsymbol{\Theta}^k}(\boldsymbol{x})\right)^2 h(\boldsymbol{x}) d\boldsymbol{x}\,.$$

To fulfill this part's goal, we have just left to find a connection between the KL divergence and the mutual information.

In the next lemma, we are interested to upper bounding the mutual information (Scarlett & Cevher, 2021) —which measures the dependence between the joint distributions and the product of the marginals of two random variables— by describing it's connection with the KL divergence. Assume that under the Markov chain $J \to f_{\boldsymbol{\Theta}^J} \to (Y^n | X^n)$, a random index $J$ is drawn uniformly from $\{1, \dots, \mathcal{M}\}$ and samples $(Y^n | X^n)$ are drawn from the prior distributions $\mathbb{P}^n_{f_{\boldsymbol{\Theta}^j}}$ corresponding to $f_{\boldsymbol{\Theta}^j} := f_{\boldsymbol{\Theta}^J}$. Note that if one sample $(Y|X)$ drawn, then we have $I(J;Y|X)$.

There are many tools to upper bounding the mutual information and the most straight forward tools is based on the KL divergence (Wainwright, 2019, Equation 15.34) as follows:

**Lemma 10 (The connection between the mutual information and the** KL divergence**)** *For any two distinct probability distributions $\mathbb{P}_{f_{\boldsymbol{\Theta}^j}}$ and $\mathbb{P}_{f_{\boldsymbol{\Theta}^k}}$ for all $j, k \in [\mathcal{M}]$, it holds that*

$$I(J;Y|X) \leq \frac{1}{\mathcal{M}^2} \sum_{\substack{j,k=1 \\ j \neq k}}^{\mathcal{M}} D_{\mathrm{KL}}\left(\mathbb{P}_{f_{\boldsymbol{\Theta}^j}} \| \mathbb{P}_{f_{\boldsymbol{\Theta}^k}}\right).$$

*For any two distinct $n$-product probability distributions $\mathbb{P}^n_{f_{\boldsymbol{\Theta}^j}}$ and $\mathbb{P}^n_{f_{\boldsymbol{\Theta}^k}}$, it holds that*

$$I(J;Y^n|X^n) \leq \frac{n}{\mathcal{M}^2} \sum_{\substack{j,k=1 \\ j \neq k}}^{\mathcal{M}} D_{\mathrm{KL}}\left(\mathbb{P}_{f_{\boldsymbol{\Theta}^j}} \| \mathbb{P}_{f_{\boldsymbol{\Theta}^k}}\right).$$

PART 2: PRELIMINARY RESULTS FOR DERIVING A LOWER BOUND FOR PACKING NUMBER OF RELU NETWORKS

In this section, we present supporting lemmas that are included in the proof of Lemma 6 for deriving the lower bound for the packing number of shallow-ReLU network function space. We start by calculating the Gaussian integrals over a half-space. Assume that $\boldsymbol{x}$ is a realization of random variable $X$ that follows the $d$-dimensional Gaussian distribution, then we say that for $k \in \{1, \dots S\}$, $\boldsymbol{b}_k^\top \boldsymbol{x} > 0$ and $\boldsymbol{b}_k^\top \boldsymbol{x} \leq 0$ are two half-spaces of hyperplane $\boldsymbol{b}_k^\top \boldsymbol{x} = 0$ for $\boldsymbol{b}_k \in \mathbb{R}^d$. We then can define the probability density function of $\boldsymbol{x}$ with mean vector $\boldsymbol{\mu}$ and the covariance matrix $\boldsymbol{\Sigma}$ as follows:

$$p(\boldsymbol{x}, \boldsymbol{\mu}, \boldsymbol{\Sigma}) = \frac{1}{(2\pi)^{d/2}\sqrt{|\boldsymbol{\Sigma}|}} \int_{\boldsymbol{x} \in \mathcal{X}} e^{\frac{-(\boldsymbol{x}-\boldsymbol{\mu})^\top \boldsymbol{\Sigma}^{-1} (\boldsymbol{x}-\boldsymbol{\mu})}{2}} d\boldsymbol{x}\,,$$

where $|\boldsymbol{\Sigma}| \equiv \det(\boldsymbol{\Sigma})$, is the determinant of $\boldsymbol{\Sigma}$.

If $\boldsymbol{\mu} = \boldsymbol{0}$, then we have

$$p(\boldsymbol{x}, \boldsymbol{0}, \boldsymbol{\Sigma}) = \frac{1}{(2\pi)^{d/2}\sqrt{|\boldsymbol{\Sigma}|}} \int_{\boldsymbol{x} \in \mathcal{X}} e^{\frac{-\boldsymbol{x}^\top \boldsymbol{\Sigma}^{-1} \boldsymbol{x}}{2}} d\boldsymbol{x}\,.$$

Accordingly, the probability density function of $\boldsymbol{x}$ on either half-space $\boldsymbol{b}_k^\top \boldsymbol{x} > 0$ or $\boldsymbol{b}_k^\top \boldsymbol{x} \leq 0$ takes the form

$$p(\boldsymbol{b}_k^\top \boldsymbol{x} > 0, \boldsymbol{0}, \boldsymbol{\Sigma}) = \frac{1}{(2\pi)^{d/2}\sqrt{|\boldsymbol{\Sigma}|}} \int_{\boldsymbol{b}_k^\top \boldsymbol{x} > \boldsymbol{0}} e^{\frac{-\boldsymbol{x}^\top \boldsymbol{\Sigma}^{-1} \boldsymbol{x}}{2}} d\boldsymbol{x}\,,$$

and can be calculated as the following lemma:

**Lemma 11 (Gaussian integrals over a half-space)** *Assume that $\boldsymbol{x}, \boldsymbol{b}_k \in \mathbb{R}^d$ and for a fixed vector $\boldsymbol{b}_k$, we define a half-space $\boldsymbol{b}_k^\top \boldsymbol{x} > 0$. Then, for the corresponding probability density function it holds that*

$$p(\boldsymbol{b}_k^\top \boldsymbol{x} > 0, \boldsymbol{0}, \boldsymbol{\Sigma}) = \frac{1}{2}\,.$$

In the next lemma we are motivated to employ the result of Lemma 11 to calculate $\mathbb{E}[(\phi(\boldsymbol{b}_k^\top \boldsymbol{x}))^2]$ which is necessary for the proof of Lemma 6.

**Lemma 12** *Let $\boldsymbol{x}$ be a Gaussian random variable and $\boldsymbol{b}_k^\top \boldsymbol{x} > 0$ is a half-space, then,*

$$\mathbb{E}\left[\left(\phi(\boldsymbol{b}_k^\top \boldsymbol{x})\right)^2\right] = \frac{1}{2}\,.$$

The aim of the next lemma is to find the joint probability density function of two uncorrelated random variables. This result is useful given that $\boldsymbol{x} \sim \mathcal{N}(0, \boldsymbol{I}_d)$ and $\boldsymbol{b}_j, \boldsymbol{b}_k \in \mathbb{R}^d$ for $j \neq k \in \{1, \ldots, S\}$, where $\boldsymbol{b}_k^\top \boldsymbol{x}, \boldsymbol{b}_j^\top \boldsymbol{x} \in \mathbb{R}$ are standard normal variables. The joint probability density function $p(\boldsymbol{b}_k^\top \boldsymbol{x} > 0 \cap \boldsymbol{b}_j^\top \boldsymbol{x} > 0)$ is calculated as presented in the following lemma:

**Lemma 13 (Joint probability density function of two uncorrelated standard normal random variables)**
*Assume that $\boldsymbol{x} \sim \mathcal{N}(\boldsymbol{0}, \boldsymbol{I}_d)$ is a random variable, then for $\boldsymbol{b}_k^\top \boldsymbol{x}$ and $\boldsymbol{b}_j^\top \boldsymbol{x}$ for all $k \neq j \in \{1, \ldots, S\}$, we can get*

$$p(\boldsymbol{b}_k^\top \boldsymbol{x} > 0 \cap \boldsymbol{b}_j^\top \boldsymbol{x} > 0) = \mathbb{E}\big[\phi(\boldsymbol{b}_k^\top \boldsymbol{x})\phi(\boldsymbol{b}_j^\top \boldsymbol{x})\big] = 0\,.$$

We use the result of this lemma in the proof of Lemma 6 to establish a lower bound for the logarithm of the packing number of a shallow-ReLU network space. In the following lemma, we present M. Klusowski & R. Barron (2017, Lemma1), which concerns the cardinality of a set and is integral to the proof of Lemma 6. This lemma helps us define our desired set with a predefined Hamming weight, and its elements can be interpreted as binary codes. Now, let state the lemma.

**Lemma 14** *For integers $d$ and $d'$ with $d \in [10, \infty)$ and $d' \in [1, d/10]$, define a set*

$$\mathcal{S} := \big\{\boldsymbol{w} \in \{0,1\}^d : \|\boldsymbol{w}\|_1 = d'\big\}\,.$$

*Then, there exists a subset $\mathcal{A} \subset \mathcal{S}$ with cardinality at least $S := \sqrt{\binom{d}{d'}}$ such that each element has Hamming weight $d'$ and any pairs of elements have minimum Hamming distance $d'/5$.*

## B   PROOFS

We begin by presenting the proof of our main theorem (Theorem 1). Subsequently, we provide the proofs for Lemma 6 and Lemma 7. Additionally, the proofs of the lemmas in Appendix A will be included.

### B.1   PROOF OF THEOREM 1

**Proof** Our objective for this proof is to establish a lower bound on the mini-max risk for ReLU neural networks. To accomplish this, we utilize a generic schema of Fano's inequality (Lemma 3) followed by the use of Lemma 4, Lemma 6 and Lemma 7. We begin by considering Fano's inequality, where, intuitively by decreasing $\delta$ sufficiently, we may ensure that

$$\frac{I(J; Y^n | X^n) + \log 2}{\log \mathcal{M}\big(2\delta, \mathcal{F}, \|\cdot\|_{L_2}\big)} \leq \frac{1}{2}\,,$$

which can be reformulated as

$$\log \mathcal{M}\big(2\delta, \mathcal{F}, \|\cdot\|_{L_2}\big) \geq 2\big(I(J; Y^n | X^n) + \log 2\big)\,.$$

Accordingly, Fano's inequality takes the form

$$\mathcal{R}_{(n,d)}(\mathcal{F}; \Phi \circ \rho) \geq \frac{1}{2}\Phi(\delta)\,.$$

Then, considering Wainwright (2019, Equation 15.13b), we can 1. use the upper bound for $I(J; Y^n | X^n)$ obtained in Lemma 4 and 2. consider the fact that $2\log 2 > 0$ to obtain

$$\log \mathcal{M}\big(2\delta, \mathcal{F}, \|\cdot\|_{L_2}\big) \geq \left(\frac{4n(\kappa\delta)^2}{\sigma^2} + 2\log 2\right)$$

$$\geq \left(\frac{4n(\kappa\delta)^2}{\sigma^2}\right).$$

If we choose $\delta$ in a way that the inequality be verified by a lower bound of the $\log \mathcal{M}(2\delta, \mathcal{F}, \|\cdot\|_{L_2})$, then, we can also make sure that it will be verified in general. Collecting the results of Lemma 6 and Lemma 7 for lower bounding the $\log \mathcal{M}(2\delta, \mathcal{F}, \|\cdot\|_{L_2})$ for ReLU-neural networks, then it holds that

$$\frac{4}{\sigma^2} n \kappa^2 \delta^2 = \left(\frac{v_1}{13\delta}\right)^2 \log(d).$$

To satisfy the lower bound for $\log \mathcal{M}(2\delta, \mathcal{F}, \|\cdot\|_{L_2})$, a suitable value for $\delta$ is

$$\delta^4 = \frac{(v_1 \sigma)^2 \log(d)}{676 n \kappa^2},$$

that implies

$$\delta = \left(\frac{(v_1 \sigma)^2 \log(d)}{676 n \kappa^2}\right)^{1/4}.$$

Substituting the obtained value of $\delta$ into Fano's inequality yields

$$\mathcal{R}_{(n,d)}(\mathcal{F}; \Phi \circ \rho) \geq \frac{1}{2} \Phi \left[ \left(\frac{(v_1 \sigma)^2 \log(d)}{676 n \kappa^2}\right)^{1/4} \right]$$

$$= \frac{1}{2} \Phi \left[ \left(\frac{v_1 \sigma}{26 \kappa}\right)^{1/2} \left(\frac{\log(d)}{n}\right)^{1/4} \right].$$

We can plug the value of $c$— in the definition of Theorem 1— into this inequality and get

$$\mathcal{R}_{(n,d)}(\mathcal{F}; \Phi \circ \rho) \geq \frac{1}{2} \Phi \left[ c \sqrt{v_1} \left(\frac{\log(d)}{n}\right)^{1/4} \right],$$

which proves our first claim.

For the second claim, we simply use $\Phi(\cdot) = (\cdot)^2$ to obtain

$$\mathcal{R}_{(n,d)}(\mathcal{F}; \Phi \circ \rho) \geq \frac{c^2}{2} v_1 \sqrt{\frac{\log(d)}{n}}.$$

Based on our mini-max risk setting (Section 2), the above expression can be presented as follows:

$$\inf_{\widehat{f}} \sup_{f^* \in \mathcal{F}} \mathbb{E}_{(\boldsymbol{x}_i, y_i)_{i=1}^n} \left[ \|\widehat{f} - f^*\|_{L_2}^2 \right] \geq \frac{c^2}{2} v_1 \sqrt{\frac{\log(d)}{n}},$$

as desired. ∎

## B.2 PROOF OF LEMMA 6

**Proof** The core of this proof involves two steps: First, the construction of a subclass of functions within function space $\mathcal{F}_{v_0, v_1}$, and then finding the lower bound for $\log$ of the cardinality of the constructed class. Second, the fact that a lower bound for the cardinality of a smaller function space can serve as a lower bound for the cardinality of the larger function space. Let us begin by discussing the construction of the subclass of the function class $\mathcal{F}_{v_0, v_1}$.

### STEP 1: CONSTRUCT A SUBCLASS OF FUNCTION CLASS $\mathcal{F}_{v_0, v_1}$

Our first step is to construct a subclass of our defined function class $\mathcal{F}_{v_0, v_1}$ and then find a lower bound for the $\log$ of the packing number of the constructed class. To achieve this, we begin by defining a set of binary vectors $\mathcal{C} \in \{0, 1\}^d$ for $d \in [10, \infty)$ such that each element of this set has a Hamming weight of $d'$, where $d' \in [1, d/10]$ and the cardinality of this set is denoted by $S$. Recall that, we assume that $v_0 = 1$, so, we can choose $d' = v_0^2 = 1$. Then, anyone can readily conclude that $S = d$ and we can consider the vector $\boldsymbol{b}_i \in \{0, 1\}^d$ as a vector with all the entries equal to zero except for the $i$th entry, which is set to one. It implies that for all $i \neq j \in \{1, \ldots, S\}$

$$|\boldsymbol{b}_i^\top \boldsymbol{b}_j| = 0.$$

We can also conclude that for each $\boldsymbol{b}_i$ with $i \in \{1, \ldots, S\}$, we have

$$\|\boldsymbol{b}_i\|_2 = \sqrt{\left((\boldsymbol{b}_i)_1\right)^2 + \ldots + \left((\boldsymbol{b}_i)_d\right)^2} = 1 \,.$$

Following the same argument as above we have

$$\|\boldsymbol{b}_i\|_1 = \left|(\boldsymbol{b}_i)_1\right| + \ldots + \left|(\boldsymbol{b}_i)_d\right| = 1 \,.$$

Then, for an enumeration $\boldsymbol{b}_1, \ldots, \boldsymbol{b}_S$ of $\mathcal{C}$, define a subclass of $\mathcal{F}_{v_0, v_1}$ by

$$\mathcal{F}_0 := \left\{ f_{(\boldsymbol{w}, \boldsymbol{b}')}(\boldsymbol{x}) := \frac{v_1}{\lambda} \sum_{k=1}^{S} w_k \phi^1(\boldsymbol{b}_k^\top \boldsymbol{x}) \ : \boldsymbol{w} \in \mathcal{A} \right\},$$

where $\boldsymbol{b}' := (v_1/\lambda)\boldsymbol{b}$. The set $\mathcal{A} := \{\boldsymbol{w} \in \{0,1\}^S \ : \|\boldsymbol{w}\|_1 = \lambda\}$ is the set in Lemma 14 and $\lambda \in [1, S/10]$ is the Hamming weight of each element of the set $\mathcal{A}$ (M. Klusowski & R. Barron, 2017, Theorem 2). According to the above definition of $\mathcal{F}_0$, we have

$$\mathbb{E}\big[\|f_{(\boldsymbol{w}, \boldsymbol{b}')}(\boldsymbol{x}) - f_{(\boldsymbol{w}', \boldsymbol{b}')}(\boldsymbol{x})\|_{L_2}^2\big] = \left(\frac{v_1}{\lambda}\right)^2 \mathbb{E}\bigg[\Big(\sum_{k=1}^{S}(w_k - w_k')\phi^1(\boldsymbol{b}_k^\top \boldsymbol{x})\Big)^2\bigg],$$

where $\boldsymbol{w}, \boldsymbol{w}' \in \mathcal{A}$.

Note that, based on the structure of $\boldsymbol{w}$ and $\boldsymbol{w}'$, for all $k \in \{1, \ldots, S\}$, $(w_k - w_k')$ falls within the set $\{-1, 0, 1\}$. And if $(w_k - w_k') = 0$, the value of the expected term on the right-hand side –for the corresponding $k$– is equal to 0; thus for the sake of convenience, we consider an integer value $S' < S$ in such a way that $|w_k - w_k'| = 1$ for all $k \in \{1, \ldots, S'\}$. Based on the structure of all pairs $\boldsymbol{w}, \boldsymbol{w}' \in \mathcal{A}$ and Lemma 14, we can conclude that $S' \geq \lambda/5$. We will use $S'$ for the remainder of the proof.

We then proceed with

$$\mathbb{E}\big[\|f_{(\boldsymbol{w}, \boldsymbol{b}')}(\boldsymbol{x}) - f_{(\boldsymbol{w}', \boldsymbol{b}')}(\boldsymbol{x})\|_{L_2}^2\big] = \left(\frac{v_1}{\lambda}\right)^2 \mathbb{E}\bigg[\Big(\sum_{k=1}^{S'}((w_k - w_k')\phi^1(\boldsymbol{b}_k^\top \boldsymbol{x}))\Big)^2\bigg]. \qquad (7)$$

Next, we are motivated to find a lower bound for $\mathbb{E}\big[\|f_{(\boldsymbol{w}, \boldsymbol{b}')}(\boldsymbol{x}) - f_{(\boldsymbol{w}', \boldsymbol{b}')}(\boldsymbol{x})\|_{L_2}^2\big]$. We can 1. employ the result of Lemma 13, which shows that $\mathbb{E}[\phi(\boldsymbol{b}_k^\top \boldsymbol{x})\phi(\boldsymbol{b}_j^\top \boldsymbol{x})] = 0$ for all distinct $j$ and $k$, to help us write the above variance over a sum, as a sum over the variance of individual entries, 2. invoke the above assumption that $(w_k - w_k')^2 = 1$, 3. use the result of Lemma 12, which shows $\mathbb{E}[(\phi(\boldsymbol{b}_k^\top \boldsymbol{x}))^2] = 1/2$ and the properties of sum, 4. apply the conclusion that $S' \geq \lambda/5$ and 5. perform some simplification to obtain

$$\mathbb{E}\big[\|f_{(\boldsymbol{w}, \boldsymbol{b}')}(\boldsymbol{x}) - f_{(\boldsymbol{w}', \boldsymbol{b}')}(\boldsymbol{x})\|_{L_2}^2\big] = \left(\frac{v_1}{\lambda}\right)^2 \sum_{k=1}^{S'} \mathbb{E}\Big[\big((w_k - w_k')\phi^1(\boldsymbol{b}_k^\top \boldsymbol{x})\big)^2\Big]$$

$$= \left(\frac{v_1}{\lambda}\right)^2 \sum_{k=1}^{S'} \mathbb{E}\Big[\big(\phi^1(\boldsymbol{b}_k^\top \boldsymbol{x})\big)^2\Big]$$

$$= \left(\frac{v_1}{\lambda}\right)^2 \frac{S'}{2}$$

$$\geq \left(\frac{v_1}{\lambda}\right)^2 \frac{\lambda}{10}$$

$$= \frac{v_1^2}{10\lambda} \,.$$

So, a $2\delta$-separation implies

$$(2\delta)^2 = \frac{v_1^2}{10\lambda} \implies \lambda = (v_1/\sqrt{40}\delta)^2 \,.$$

Then, we can 1. use the result of Lemma 14 that $\log(\#\mathcal{F}_0)$ denotes as the log of the cardinality of $\mathcal{F}_0$ is at least $\log \binom{S}{\lambda} \geq (\lambda/4)\log(S)$, 2. plugin the value of $S$, 3. use the fact that $\sqrt{169} > \sqrt{160}$

and 4. perform some simplification that gives

$$\log(\#\mathcal{F}_0) \geq \Big(\frac{v_1}{\sqrt{160\delta}}\Big)^2 \log(S)$$

$$= \Big(\frac{v_1}{\sqrt{160\delta}}\Big)^2 \log(d)$$

$$\geq \Big(\frac{v_1}{\sqrt{169\delta}}\Big)^2 \log(d)$$

$$= \Big(\frac{v_1}{13\delta}\Big)^2 \log(d).$$

Based on the formula $\lambda = (v_1/\sqrt{40\delta})^2$, when $v_1$ is fixed, it is evident that as $\delta$ decreases, $\lambda$ increases. Moreover, since $\lambda \leq d/10$, we need to assume that $d$ is large enough. In particular, the logarithmic dependence on the input dimension $d$, offers a perspective on how network growth relates to the dimensionality of the problem at hand.

STEP 2: DERIVING A LOWER BOUND FOR $\log(\#\mathcal{F}_{v_0,v_1})$

For the second step, our aim is to lower bound the $\log$ of the cardinality of the function class $\mathcal{F}_{v_0,v_1}$ using the result of the first step. Since we define $\mathcal{F}_0$ as a subclass of $\mathcal{F}_{v_0,v_1}$, we can conclude that the lower bound established for $\log(\#\mathcal{F}_0)$ in the first step also serves as a lower bound for $\log(\#\mathcal{F}_{v_0,v_1})$. We then can get

$$\log \mathcal{M}\big(2\delta, \mathcal{F}_{v_0,v_1}, \|\cdot\|_{L_2}\big) \geq \Big(\frac{v_1}{13\delta}\Big)^2 \log(d),$$

as desired. ■

### B.3  PROOF OF LEMMA 7

**Proof**  We claim that a deep-ReLU network can generate a shallow-ReLU network and the idea is based on Hebiri & Lederer (2020, Theorem 1). The idea is as follows: For any two consecutive layers $j$ and $j-1$, we can redefine a network of depth $L$ as a network of $L-1$ through a merged weight of these two layers and a merged activation function. Motivated by this idea, we first apply it for a two-hidden-layer neural network. For a two-hidden-layer neural network $f_{\boldsymbol{\Theta}}(\boldsymbol{x}) := W^2\boldsymbol{\phi}^2[W^1\boldsymbol{\phi}^1[W^0\boldsymbol{x}]]$ with $W^1 \geq 0$ (by $W^1 \geq 0$ we mean all coordinates of $W^1$ are non-negative), it holds that

$$W^2\boldsymbol{\phi}^2\big[W^1\boldsymbol{\phi}^1[W^0\boldsymbol{x}]\big] = W^2\boldsymbol{\phi}^{2,1}[W^{1,0}\boldsymbol{x}],$$

where $W^{1,0} = W^1W^0$ and $\boldsymbol{\phi}^{2,1}$ as the merged activation functions $\boldsymbol{\phi}^2$ and $\boldsymbol{\phi}^1$. For the ease of comparison, define a shallow-ReLU network with different parameters as follows:

$$f_{\mathrm{Sh}}[\boldsymbol{x}] := \boldsymbol{\gamma}\boldsymbol{\phi}[\psi\boldsymbol{x}].$$

It basically means that for generating a shallow-ReLU network $f_{\mathrm{Sh}}$ using a two-hidden-layer ReLU network, all we need to do is, decomposing the inner layer of a shallow-ReLU network in a way that

$$\boldsymbol{\gamma}\boldsymbol{\phi}[\psi\boldsymbol{x}] = \boldsymbol{\gamma}\boldsymbol{\phi}^2\big[W^1\boldsymbol{\phi}^1[W^0\boldsymbol{x}]\big],$$

where $W^1 \geq 0$. To be sure that $W^1$ is a non-negative matrix, we consider $W^1 = \boldsymbol{I}_\omega$ (recall that we consider neural networks with equal widths for all hidden layers denoted as $\omega$). Now, we can 1. use the non-negativity of $W^1$, 2. employ the merged activation function's property, 3. define $\psi := W^1W^0$ and 4. define $\boldsymbol{\gamma} := W^2$ to get

$$W^2\boldsymbol{\phi}^2\big[W^1\boldsymbol{\phi}^1[W^0\boldsymbol{x}]\big] = W^2\boldsymbol{\phi}^2\big[\boldsymbol{\phi}^1[W^1W^0\boldsymbol{x}]\big]$$

$$= W^2\boldsymbol{\phi}^{2,1}[W^1W^0\boldsymbol{x}]$$

$$= W^2\boldsymbol{\phi}^{2,1}[\psi\boldsymbol{x}]$$

$$= \boldsymbol{\gamma}\boldsymbol{\phi}^{2,1}[\psi\boldsymbol{x}],$$

where $\phi := \phi^{2,1}$. To establish that the equivalence also extends to deep-ReLU networks with $L > 2$ and $W^i \geq 0$ for all $i \in [1, (L-1)]$, we employ a similar approach as applied in the case of two-hidden-layer-ReLU networks. We begin with a deep-ReLU network with $L$ hidden layers and proceed by iteratively reducing the network's depth by one layer in each step. By repeating this process $(L-1)$ times, we consequently derive

$$
\begin{aligned}
f_{\boldsymbol{\Theta}}(\boldsymbol{x}) &= W^L \boldsymbol{\phi}^L \Big[ W^{L-1} \boldsymbol{\phi}^{L-1} \big[ \ldots W^1 \boldsymbol{\phi}^1 [W^0 \boldsymbol{x}] \big] \Big] \\
&= W^L \boldsymbol{\phi}^L \Big[ W^{L-1} \boldsymbol{\phi}^{L-1} \big[ \cdots W^2 \boldsymbol{\phi}^{2,1} [W^{1,0} \boldsymbol{x}] \big] \Big] \\
&= \ldots \\
&= W^L \boldsymbol{\phi}^{L,\cdots,1} [W^{L-1,\cdots,0} \boldsymbol{x}] \\
&= \boldsymbol{\gamma} \boldsymbol{\phi} [\psi \boldsymbol{x}] \,,
\end{aligned}
$$

where $\boldsymbol{\phi} := \phi^{L,\cdots,1} = \phi^L \phi^{L-1} \ldots \phi^1$ is a merged activation function, $\boldsymbol{\gamma} := W^L$ and $\psi := W^{L-1,\cdots,0} = W^{L-1} W^{L-2} \cdots W^0$, where $W^1 = W^2 = \cdots = W^{L-1} = \boldsymbol{I}_\omega$.

Assume that we have a ReLU neural network function space $\mathcal{F}_{\mathcal{B}_{\mathrm{L}}}$ including the networks with $L$ hidden-layers and with width $\omega$ and the corresponding network parameters $\mathcal{B}_{\mathrm{L}}$. Then, we claim that such the network space can behave like a shallow-ReLU network function space $\mathcal{F}_{\mathcal{B}_{\mathrm{Sh}}}$ with parameters $\mathcal{B}_{\mathrm{Sh}}$. That means, a shallow-ReLU network space parameterized by $\mathcal{B}_{\mathrm{Sh}}$ is a subset of the network space parameterized by $\mathcal{B}_{\mathrm{L}}$. In other words

$$ \mathcal{F}_{\mathcal{B}_{\mathrm{Sh}}} \subset \mathcal{F}_{\mathcal{B}_{\mathrm{L}}} \,, $$

as desired.

For our second claim, we know that for the common packing set (in our case $2\delta$ separated set), the packing number of a space would be proportional to its size. Thus, using the the view of the first claim, we can conclude that a lower bound for the packing number of $\mathcal{F}_{\mathcal{B}_{\mathrm{Sh}}}$ can also be served as a lower bound for the packing number of function space $\mathcal{F}_{\mathcal{B}_{\mathrm{L}}}$ which gives $\log \mathcal{M}(2\delta, \mathcal{F}_{\mathcal{B}_{\mathrm{Sh}}}, \|\cdot\|_{L_2}) \leq \log \mathcal{M}(2\delta, \mathcal{F}_{\mathcal{B}_{\mathrm{L}}}, \|\cdot\|_{L_2})$. So, we can use our results in Lemma 6 to obtain that

$$ \log \mathcal{M}\big(2\delta, \mathcal{F}_{\mathcal{B}_{\mathrm{L}}}, \|\cdot\|_{L_2}\big) \geq \log \mathcal{M}\big(2\delta, \mathcal{F}_{\mathcal{B}_{\mathrm{Sh}}}, \|\cdot\|_{L_2}\big) \geq \left(\frac{v_1}{13\delta}\right)^2 \log(d) \,, $$

as desired. ∎

**Remark 15 (Compatibility with Leaky ReLU networks)** *According to the framework specified in Hebiri & Lederer (2020) for activation functions, the first claim ($\mathcal{F}_{\mathcal{B}_{\mathrm{Sh}}} \subset \mathcal{F}_{\mathcal{B}_{\mathrm{L}}}$) holds true for leaky ReLU networks as well.*

**Remark 16 (He initialization for weight parameters' scaling)** *We claim that our weight parameters' scaling is based on He weight initialization (He et al., 2015). He initialization method is calculated as a random number with a Gaussian probability distribution with a mean of $0$ and a standard deviation of $(\sqrt{2/m})$, where $m$ is the number of inputs to the node. In our deep network setting with $L$ hidden layers, it is assumed that $W^1 = W^2 = \ldots = W^{L-1} = \boldsymbol{I}_\omega$, where $\omega$ represents the width of a shallow network (a subset of the shallow network function space $\mathcal{F}_{\mathcal{B}_{\mathrm{Sh}}}$), that is also equal to the number of hidden nodes in each hidden layer. Based on this setting, it can be readily concluded that $\|W^1\|_1 = \|W^2\|_1 = \ldots = \|W^{L-1}\|_1 = \|\boldsymbol{I}_\omega\|_1 = \omega$. Furthermore, by using He initialization, we can achieve the following result:*

$$
\begin{aligned}
\|W^1\|_1 = \|W^2\|_1 = \ldots = \|W^{L-1}\|_1 &\sim \frac{\sqrt{2}\omega^2}{\sqrt{\omega}} \\
&= \sqrt{2}(\omega^{3/2}) \,.
\end{aligned}
$$

### B.4 PROOF OF LEMMA 9

**Proof** To calculate the KL divergence between two normal distributions $\mathbb{P}_{f_{\boldsymbol{\Theta}^j}}$ and $\mathbb{P}_{f_{\boldsymbol{\Theta}^k}}$ of a continuous random variable, each with the corresponding densities $p_{f_{\boldsymbol{\Theta}^j}}(\boldsymbol{z})$ and $p_{f_{\boldsymbol{\Theta}^k}}(\boldsymbol{z})$,

for all $j, k \in [\mathcal{M}]$ where $j \neq k$, we can 1. use the definition of the KL divergence, 2. plug the value of $p_{f_{\Theta^j}}(\boldsymbol{z})$ and $p_{f_{\Theta^k}}(\boldsymbol{z})$ in, 3. perform some simplification, 4. apply the definition of expected value, 5. the linearity of expectation, 6. use $y = f_{\Theta^j}(\boldsymbol{x}) + u$, 7. perform further rewriting, 8. apply the linearity of expected value, assuming independence between each $u_i$ and $\boldsymbol{x}_i$, 9. cancel out the second term ($\mathbb{E}[u] = 0$) and 10. recognize that only $\boldsymbol{x}$ values remain, to get

$$
\begin{aligned}
D_{\mathrm{KL}}(\mathbb{P}_{f_{\Theta^j}} \,||\, \mathbb{P}_{f_{\Theta^k}}) &= \int_{\mathcal{X} \times \mathcal{Y}} p_{f_{\Theta^j}}(\boldsymbol{z}) \log \frac{p_{f_{\Theta^j}}(\boldsymbol{z})}{p_{f_{\Theta^k}}(\boldsymbol{z})} d\boldsymbol{z} \\
&= \int_{\mathcal{X} \times \mathcal{Y}} p_{f_{\Theta^j}}(\boldsymbol{z}) \log \left( \frac{(1/\sqrt{2\pi\sigma^2}) e^{-\left((y - f_{\Theta^j}(\boldsymbol{x}))^2/2\sigma^2\right)} h(\boldsymbol{x})}{(1/\sqrt{2\pi\sigma^2}) e^{-\left((y - f_{\Theta^k}(\boldsymbol{x}))^2/2\sigma^2\right)} h(\boldsymbol{x})} \right) d\boldsymbol{z} \\
&= \int_{\mathcal{X} \times \mathcal{Y}} p_{f_{\Theta^j}}(\boldsymbol{z}) \frac{1}{2\sigma^2} \left( (y - f_{\Theta^k}(\boldsymbol{x}))^2 - (y - f_{\Theta^j}(\boldsymbol{x}))^2 \right) d\boldsymbol{z} \\
&= \mathbb{E}_{\boldsymbol{z} \sim p_{f_{\Theta^j}}(\boldsymbol{z})} \left[ \frac{1}{2\sigma^2} \left( (y - f_{\Theta^k}(\boldsymbol{x}))^2 - (y - f_{\Theta^j}(\boldsymbol{x}))^2 \right) \right] \\
&= \frac{1}{2\sigma^2} \mathbb{E}_{\boldsymbol{z} \sim p_{f_{\Theta^j}}(\boldsymbol{z})} \left[ (y - f_{\Theta^k}(\boldsymbol{x}))^2 - (y - f_{\Theta^j}(\boldsymbol{x}))^2 \right] \\
&= \frac{1}{2\sigma^2} \mathbb{E}_{\boldsymbol{z} \sim p_{f_{\Theta^j}}(\boldsymbol{z})} \left[ (f_{\Theta^j}(\boldsymbol{x}) + u - f_{\Theta^k}(\boldsymbol{x}))^2 - (u)^2 \right] \\
&= \frac{1}{2\sigma^2} \mathbb{E}_{\boldsymbol{z} \sim p_{f_{\Theta^j}}(\boldsymbol{z})} \left[ (f_{\Theta^j}(\boldsymbol{x}) - f_{\Theta^k}(\boldsymbol{x}))^2 - 2u(f_{\Theta^j}(\boldsymbol{x}) - f_{\Theta^k}(\boldsymbol{x})) \right] \\
&= \frac{1}{2\sigma^2} \left( \mathbb{E}_{\boldsymbol{z} \sim p_{f_{\Theta^j}}(\boldsymbol{z})} \left[ (f_{\Theta^j}(\boldsymbol{x}) - f_{\Theta^k}(\boldsymbol{x}))^2 \right] - 2\mathbb{E}[u] \mathbb{E}_{\boldsymbol{z} \sim p_{f_{\Theta^j}}(\boldsymbol{z})} \left[ f_{\Theta^j}(\boldsymbol{x}) - f_{\Theta^k}(\boldsymbol{x}) \right] \right) \\
&= \frac{1}{2\sigma^2} \left( \mathbb{E}_{\boldsymbol{z} \sim p_{f_{\Theta^j}}(\boldsymbol{z})} \left[ (f_{\Theta^j}(\boldsymbol{x}) - f_{\Theta^k}(\boldsymbol{x}))^2 \right] \right) \\
&= \frac{1}{2\sigma^2} \int_{\boldsymbol{x} \in \mathcal{X}} (f_{\Theta^j}(\boldsymbol{x}) - f_{\Theta^k}(\boldsymbol{x}))^2 h(\boldsymbol{x}) d\boldsymbol{x} \,.
\end{aligned}
$$

Furthermore, by combining this result with Lemma 8's result, it holds that for all $j, k \in [\mathcal{M}]$ and $j \neq k$

$$
D_{\mathrm{KL}}(\mathbb{P}_{f_{\Theta^j}}^n \,||\, \mathbb{P}_{f_{\Theta^k}}^n) = \frac{n}{2\sigma^2} \int_{\boldsymbol{x} \in \mathcal{X}} (f_{\Theta^j}(\boldsymbol{x}) - f_{\Theta^k}(\boldsymbol{x}))^2 h(\boldsymbol{x}) d\boldsymbol{x} \,,
$$

as desired. ∎

### B.5 PROOF OF LEMMA 10

**Proof** Consider a family of distributions $\{\mathbb{P}_{f_{\Theta^1}}, \ldots, \mathbb{P}_{f_{\Theta^{\mathcal{M}}}}\}$, then $I(J; Y|X)$ with respect to $J \to f_{\Theta^J} \to (Y|X)$, can be defined by using the KL divergence —as the underlying measure of distance— (Wainwright, 2019, Equation 15.29)

$$
I(J; Y|X) := D_{\mathrm{KL}}\big(\mathbb{Q}_{(X,Y),J} \,||\, \mathbb{Q}_{(X,Y)} \mathbb{Q}_J\big) \,,
$$

where $\mathbb{Q}_{(X,Y),J}$ is the joint distribution of the pair $((X, Y), J)$ and $\mathbb{Q}_{(X,Y)} \mathbb{Q}_J$ is the product of their marginals, and assume that $\overline{\mathbb{Q}} \equiv \mathbb{Q}_{(X,Y)} := 1/\mathcal{M} \sum_{j=1}^{\mathcal{M}} \mathbb{P}_{f_{\Theta^j}}$ is the mixture distribution. Then, $I(J; Y|X)$ can be written in terms of component distributions $\{\mathbb{P}_{f_{\Theta^j}}, j \in [\mathcal{M}]\}$ as follows:

$$
I(J; Y|X) = \frac{1}{\mathcal{M}} \sum_{j=1}^{\mathcal{M}} D_{\mathrm{KL}}\big(\mathbb{P}_{f_{\Theta^j}} \,||\, \overline{\mathbb{Q}}\big) \,.
$$

Intuitively, it means the mean the KL divergence between $\mathbb{P}_{f_{\Theta^j}}$ and $\overline{\mathbb{Q}}$- averaged over the choice of index $j$- gives the mutual information. Furthermore, based on the definition of the KL divergence, we can conclude that for $j = k$

$$
D_{\mathrm{KL}}\big(\mathbb{P}_{f_{\Theta^j}} \,||\, \mathbb{P}_{f_{\Theta^k}}\big) = 0 \,.
$$

Accordingly, we can 1. employ the mixture distribution formula in the above equation, 2. use the convexity of the KL divergence and apply Jensen inequality and 3. use the linearity property of sum to obtain

$$
\begin{aligned}
I(J;Y|X) &= \frac{1}{\mathcal{M}} \sum_{j=1}^{\mathcal{M}} D_{\mathrm{KL}}\left( \mathbb{P}_{f_{\Theta^j}} \,\|\, \frac{1}{\mathcal{M}} \sum_{k=1}^{\mathcal{M}} \mathbb{P}_{f_{\Theta^k}} \right) \\
&\leq \frac{1}{\mathcal{M}} \left( \sum_{j=1}^{\mathcal{M}} \left( \frac{1}{\mathcal{M}} \sum_{k=1}^{\mathcal{M}} D_{\mathrm{KL}}\left( \mathbb{P}_{f_{\Theta^j}} \,\|\, \mathbb{P}_{f_{\Theta^k}} \right) \right) \right) \\
&= \frac{1}{\mathcal{M}^2} \sum_{\substack{j,k=1 \\ j\neq k}}^{\mathcal{M}} D_{\mathrm{KL}}\left( \mathbb{P}_{f_{\Theta^j}} \,\|\, \mathbb{P}_{f_{\Theta^k}} \right).
\end{aligned}
$$

Consequently, if we can construct a $2\delta$-packing set such that all two distinct pairs of distributions $\mathbb{P}_{f_{\Theta^j}}$ and $\mathbb{P}_{f_{\Theta^k}}$ are close in average, then the mutual information can be controlled.

For the second claim, we employ the previous view with the result of Lemma 8 to get

$$
I(J;Y^n|X^n) \leq \frac{n}{\mathcal{M}^2} \sum_{\substack{j,k=1 \\ j\neq k}}^{\mathcal{M}} D_{\mathrm{KL}}\left( \mathbb{P}_{f_{\Theta^j}} \,\|\, \mathbb{P}_{f_{\Theta^k}} \right),
$$

as desired. ∎

### B.6 PROOF OF LEMMA 11

**Proof** In this proof, we first define a rotation matrix $\boldsymbol{R} \in \mathrm{SO}(d)$, which belongs to the special orthogonal group. Then, based on the fact that $\boldsymbol{R}^\top = \boldsymbol{R}^{-1}$, we can write $\boldsymbol{b}_k^\top \boldsymbol{x} = \boldsymbol{b}_k^\top \boldsymbol{R}^{-1} \boldsymbol{R} \boldsymbol{x} = (\boldsymbol{R}\boldsymbol{b}_k)^\top \boldsymbol{R}\boldsymbol{x}$. Accordingly, we can obtain

$$
p(\boldsymbol{b}_k^\top \boldsymbol{x} > 0, \boldsymbol{0}, \boldsymbol{\Sigma}) = \frac{1}{(2\pi)^{d/2}\sqrt{|\boldsymbol{\Sigma}|}} \int_{(\boldsymbol{R}\boldsymbol{b}_k)^\top \boldsymbol{R}\boldsymbol{x} > 0} e^{\frac{-(\boldsymbol{R}\boldsymbol{x})^\top \boldsymbol{R}\boldsymbol{\Sigma}^{-1}\boldsymbol{R}^\top (\boldsymbol{R}\boldsymbol{x})}{2}} d\boldsymbol{x}.
$$

By defining $\boldsymbol{G} := \boldsymbol{R}\boldsymbol{x}$ and $\tilde{\boldsymbol{b}}_k := \boldsymbol{R}\boldsymbol{b}_k$ and $d\boldsymbol{x} = (d\boldsymbol{x}/d\boldsymbol{G}) \times d\boldsymbol{G}$, we get

$$
p(\boldsymbol{b}_k^\top \boldsymbol{x} > 0, \boldsymbol{0}, \boldsymbol{\Sigma}) = \frac{1}{(2\pi)^{d/2}\sqrt{|\boldsymbol{\Sigma}|}} \int_{\tilde{\boldsymbol{b}}_k^\top \boldsymbol{G} > \boldsymbol{0}} e^{\frac{-\boldsymbol{G}^\top \boldsymbol{R}\boldsymbol{\Sigma}^{-1}\boldsymbol{R}^\top \boldsymbol{G}}{2}} \left( \frac{d\boldsymbol{x}}{d\boldsymbol{G}} \right) \times d\boldsymbol{G}.
$$

Then, 1. by setting $\tilde{\boldsymbol{\Sigma}} := \boldsymbol{R}\boldsymbol{\Sigma}^{-1}\boldsymbol{R}^\top$, $(d\boldsymbol{x}/d\boldsymbol{G}) := |\boldsymbol{R}|$ ($|\boldsymbol{R}| \equiv \det(\boldsymbol{R})$) and $\tilde{\boldsymbol{b}}_k = (\|\boldsymbol{b}_k\|_2, 0, \ldots, 0)^\top$, 2. by factoring out the term $|\boldsymbol{R}|$, 3. the fact that the probability density function of a Gaussian distribution for a random variable across its domain is 1 and 4. by noting that $|\boldsymbol{R}| = 1$, we can obtain

$$
\begin{aligned}
p(\boldsymbol{b}_k^\top \boldsymbol{x} > 0, \boldsymbol{0}, \boldsymbol{\Sigma}) &= \frac{1}{(2\pi)^{d/2}\sqrt{|\boldsymbol{\Sigma}|}} \int_{\tilde{\boldsymbol{b}}_k^\top \boldsymbol{G} > \boldsymbol{0}} e^{\frac{-\boldsymbol{G}^\top \tilde{\boldsymbol{\Sigma}}^{-1}\boldsymbol{G}}{2}} |\boldsymbol{R}| d\boldsymbol{G} \\
&= \frac{|\boldsymbol{R}|}{(2\pi)^{d/2}\sqrt{|\boldsymbol{\Sigma}|}} \int_{\tilde{\boldsymbol{b}}_k^\top \boldsymbol{G} > \boldsymbol{0}} e^{\frac{-\boldsymbol{G}^\top \tilde{\boldsymbol{\Sigma}}^{-1}\boldsymbol{G}}{2}} d\boldsymbol{G} \\
&= \frac{|\boldsymbol{R}|}{2} \\
&= \frac{1}{2},
\end{aligned}
$$

as desired. ∎

## B.7 PROOF OF LEMMA 12

**Proof** In this proof, our objective is to compute $\mathbb{E}[\phi(\boldsymbol{b}_k^\top \boldsymbol{x})\phi(\boldsymbol{b}_k^\top \boldsymbol{x})]$, which is a crucial component of the proof presented in Lemma 6, helping us establish a lower bound for a shallow-ReLU neural network. To achieve this, we can 1. employ the definition of expected value, 2. apply the definition of ReLU function, 3. perform some rewriting, 4. take out $\boldsymbol{b}_k^\top$ and $\boldsymbol{b}_k$, 5. exploit the symmetry of $\boldsymbol{x}$ (Lemma 11), 6. employ the definition of expectation, 7. apply the fact that $\mathbb{E}[\boldsymbol{x}\boldsymbol{x}^\top] = \boldsymbol{I}_d$, 8. apply $\boldsymbol{b}_k^\top \boldsymbol{I}_d \, \boldsymbol{b}_k = \boldsymbol{b}_k^\top \boldsymbol{b}_k$ and 9. use $\boldsymbol{b}_k^\top \boldsymbol{b}_k = 1$ to obtain

$$
\begin{aligned}
\mathbb{E}\big[\phi(\boldsymbol{b}_k^\top \boldsymbol{x})\phi(\boldsymbol{b}_k^\top \boldsymbol{x})\big] &= \int_{\mathcal{X}} \phi(\boldsymbol{b}_k^\top \boldsymbol{x})\phi(\boldsymbol{b}_k^\top \boldsymbol{x})h(\boldsymbol{x})d\boldsymbol{x} \\
&= \int_{\boldsymbol{x}:\boldsymbol{b}_k^\top \boldsymbol{x}>0} (\boldsymbol{b}_k^\top \boldsymbol{x})^2 h(\boldsymbol{x})d\boldsymbol{x} \\
&= \int_{\boldsymbol{x}:\boldsymbol{b}_k^\top \boldsymbol{x}>0} (\boldsymbol{b}_k^\top \boldsymbol{x})(\boldsymbol{b}_k^\top \boldsymbol{x})^\top h(\boldsymbol{x})d\boldsymbol{x} \\
&= \boldsymbol{b}_k^\top \left( \int_{\boldsymbol{x}:\boldsymbol{b}_k^\top \boldsymbol{x}>0} \boldsymbol{x}\boldsymbol{x}^\top h(\boldsymbol{x})d\boldsymbol{x} \right) \boldsymbol{b}_k \\
&= \boldsymbol{b}_k^\top \left( \frac{\int_{\mathcal{X}} \boldsymbol{x}\boldsymbol{x}^\top h(\boldsymbol{x})d\boldsymbol{x}}{2} \right) \boldsymbol{b}_k \\
&= \boldsymbol{b}_k^\top \frac{\mathbb{E}[\boldsymbol{x}\boldsymbol{x}^\top]}{2} \boldsymbol{b}_k \\
&= \frac{1}{2}\boldsymbol{b}_k^\top \boldsymbol{I}_d \, \boldsymbol{b}_k \\
&= \frac{1}{2}\boldsymbol{b}_k^\top \boldsymbol{b}_k \\
&= \frac{1}{2},
\end{aligned}
$$

as desired. ∎

## B.8 PROOF OF LEMMA 13

**Proof** Following the same approach as in the proof of Lemma 12, we can compute the term $\mathbb{E}[\phi(\boldsymbol{b}_k^\top \boldsymbol{x})\phi(\boldsymbol{b}_j^\top \boldsymbol{x})]$ that is used in the proof of Lemma 6. To do this, we can 1. use the definition of expected value, 2. note that $(\boldsymbol{b}_j^\top \boldsymbol{x}) = (\boldsymbol{b}_j^\top \boldsymbol{x})^\top$ since $\boldsymbol{b}_j^\top \boldsymbol{x} \in \mathbb{R}$, 3. perform some simplification, 4. take out the terms $\boldsymbol{b}_k^\top$ and $\boldsymbol{b}_j$ from both sides of the integral, 5. invoke the fact that $(\boldsymbol{b}_k^\top \boldsymbol{x} > 0 \cap \boldsymbol{b}_j^\top \boldsymbol{x} > 0) \subset \boldsymbol{x}$, 6. apply the definition of expected value, 7. use the property

$\mathbb{E}[\boldsymbol{x}\boldsymbol{x}^\top] = \boldsymbol{I}_d$ and 8. incorporate the fact that $\boldsymbol{b}_k^\top \boldsymbol{b}_j = 0$ to obtain

$$
\begin{aligned}
\mathbb{E}\big[\phi(\boldsymbol{b}_k^\top \boldsymbol{x})\phi(\boldsymbol{b}_j^\top \boldsymbol{x})\big] &= \int_{\mathcal{X}} \phi(\boldsymbol{b}_k^\top \boldsymbol{x})\phi(\boldsymbol{b}_j^\top \boldsymbol{x})h(\boldsymbol{x})d\boldsymbol{x} \\
&= \int_{\substack{\boldsymbol{b}_k^\top \boldsymbol{x} > 0 \\ \boldsymbol{b}_j^\top \boldsymbol{x} > 0}} \big(\boldsymbol{b}_k^\top \boldsymbol{x}\big)\big(\boldsymbol{b}_j^\top \boldsymbol{x}\big)^\top h(\boldsymbol{x})d\boldsymbol{x} \\
&= \int_{\substack{\boldsymbol{b}_k^\top \boldsymbol{x} > 0 \\ \boldsymbol{b}_j^\top \boldsymbol{x} > 0}} \big(\boldsymbol{b}_k^\top \boldsymbol{x}\boldsymbol{x}^\top \boldsymbol{b}_j\big) h(\boldsymbol{x})d\boldsymbol{x} \\
&= \boldsymbol{b}_k^\top \left( \int_{\substack{\boldsymbol{b}_k^\top \boldsymbol{x} > 0 \\ \boldsymbol{b}_j^\top \boldsymbol{x} > 0}} \big(\boldsymbol{x}\boldsymbol{x}^\top\big) h(\boldsymbol{x})d\boldsymbol{x} \right) \boldsymbol{b}_j \\
&\leq \boldsymbol{b}_k^\top \left( \int_{\mathcal{X}} \big(\boldsymbol{x}\boldsymbol{x}^\top\big) h(\boldsymbol{x})d\boldsymbol{x} \right) \boldsymbol{b}_j \\
&= \boldsymbol{b}_k^\top \mathbb{E}[\boldsymbol{x}\boldsymbol{x}^\top]\boldsymbol{b}_j \\
&= \boldsymbol{b}_k^\top \boldsymbol{I}_d \boldsymbol{b}_j \\
&= 0 .
\end{aligned}
$$

According to the fact that the expected value of the product of two non-negative terms is always non-negative, we can conclude that

$$
\mathbb{E}\big[\phi(\boldsymbol{b}_k^\top \boldsymbol{x})\phi(\boldsymbol{b}_j^\top \boldsymbol{x})\big] = 0 ,
$$

as desired. ∎

## C    EMPIRICAL DETAILS

Here, we first explain the two loss functions employed for training the network in both classification and regression datasets. Then, we provide the values for the network's hyper-parameters and the parameters of those two curves corresponding to each dataset in Table 1.

### C.1    LOSS FUNCTIONS:

In the training procedure of the deep-ReLU networks, we employ two different loss functions implemented in PyTorch: "nn.MSELoss()" for regression and "nn.CrossEntropyLoss()" for classification. Details of these loss functions are as follows:

**Cross-entropy loss**    For classification purpose, we use (categorical) Cross-entropy and define it as (Murphy, 2012; Ho & Wooky, 2019)

$$
\ell_{\mathrm{CE}}\left(f_{\boldsymbol{\Theta}}\right) := -\frac{1}{n} \sum_{k=0}^{m-1} \sum_{i=0}^{n-1} \Big( (y_i)_k \log p\big(f_{\boldsymbol{\Theta}}(\boldsymbol{x}_i), k\big) \Big) ,
$$

where $(y_i)_k$ is the $k$-th element of the one-hot vector of the target label for the $i$-th data sample and

$$
p(f_{\boldsymbol{\Theta}}(\boldsymbol{x}), k) := \frac{e^{(f_{\boldsymbol{\Theta}}(\boldsymbol{x}))_k}}{\sum_{i=0}^{m-1} e^{(f_{\boldsymbol{\Theta}}(\boldsymbol{x}))_i}} ,
$$

where $(f_{\boldsymbol{\Theta}}(\boldsymbol{x}))_k$ is the $k$-th output of a network indexed by $\boldsymbol{\Theta}$.

**Mean-squared error**    For regression, we use Mean-squared (Botchkarev, 2018) and define it as

$$
\ell_{\mathrm{MS}}\left(f_{\boldsymbol{\Theta}}\right) := \frac{1}{n} \sum_{i=1}^{n} \sum_{k=0}^{m-1} \frac{\big((y_i)_k - \big(f_{\boldsymbol{\Theta}}(\boldsymbol{x})\big)_k\big)^2}{m} ,
$$

where $(y_i)_k$, is the $k$-th element of the one-hot vector of the target label and $(f_{\boldsymbol{\Theta}}(\boldsymbol{x}))_k$ is the $k$-th output of a network indexed by $\boldsymbol{\Theta}$.

Table 1: Two curves' parameters and the ReLU neural network's hyper-parameters for $MNIST$, $CHP$ and $CIFAR - 10$ datasets.

| Parameters | $MNIST$ | $MNIST$ | $MNIST$ | $MNIST$ | $CHP$ | CIFAR | CIFAR |
|---|---|---|---|---|---|---|---|
| $c_1$ | 1.002e-15 | 7.444e-01 | 2.145e-01 | 1.456e-15 | 5.340e-02 | 1.826e+00 | 1.657e+00 |
| $\alpha$ | 3.339e+01 | 8.900e+01 | 10.436e+01 | 8.756e+01 | 1.092e+00 | 2.175e+01 | 2.488e+02 |
| $c_2$ | 1.387e-01 | 1.191e+00 | 7.370e-01 | 3.381e-01 | 8.403e-02 | 1.941e+00 | 1.797e+00 |
| $\beta$ | 1.370e+03 | 2.660e+03 | 3.340e+03 | 3.618e+03 | 5.006e+00 | 6.90e+02 | 7.954e+02 |
| $\omega$ | 9.000e+02 | 5 | 10 | 20 | 2.300e+01 | 100 | 120 |
| LR | 1.000e-03 | 1.000e-03 | 1.000e-03 | 1.000e-03 | 1.000e-02 | 1.000e-03 | 1.000e-03 |
| $L$ | 4 | 1 | 1 | 1 | 5 | 1 | 1 |

## C.2 THE STRUCTURE OF THE NEURAL NETWORK AND TWO CURVES' COEFFICIENTS

The network hyper-parameters and the parameters of the curves for each dataset are provided in Table 1.

