# OpenReview forum: "How many samples are needed to train a deep-ReLU neural network?"
_ICLR.cc/2024/Conference — Submitted to ICLR 2024_

### Official Review · Reviewer_xTTm · 2023-10-25

**Soundness:** 2 fair
**Presentation:** 2 fair
**Contribution:** 1 poor
**Rating:** 3
**Confidence:** 4

**Summary:**

In this paper, the authors propose that the generalization error of neural networks scales with a rate of 1/\sqrt{n}. Their theoretical analysis establishes a mini-max risk by assuming bounded parameter norms, while empirical evidence supports the 1/\sqrt{n} rate in MNIST and CHP datasets.

While the paper addresses an important question related to the scaling law, I find the theoretical evidence somewhat lacking. The main point of the paper is not entirely clear, and the authors need to articulate their primary contribution better. Moreover, there is limited discussion of the technical challenges and the extension of previous work to ReLU networks. (See limitation part for more details.)

Therefore, I cannot recommend acceptance for the time being.

**Strengths:**

1. The authors establish a mini-max bound for ReLU neural networks, which appears to be valid.
2. While the main contribution needs further clarification, the topic itself is interesting.

**Weaknesses:**

My major concern is that it seems that the justification of this paper is not enough.
I am still confused about which point is the main point that the authors want to show.
There are many alternatives, but they did not convince me.
For example, the technical contributions? The authors claim that previous works cannot be applied to ReLU networks, but I am not sure whether this extension has many technical difficulties.
Another one is that the theory can match the practice, but I believe that the empirical observations in this paper are not enough to support this (with only the MNIST and CHP datasets).
So I suggest that the authors could first tell me what the *major contribution* is and then convince me that the major contribution is significant.
I would increase my score then.

For the theory part:
1. The authors assume a bounded weight norm. However, this is not true. Some existing papers [e.g., uniform convergence may be unable to example generalization in deep learning] have pointed out that the weight norm may increase with n, which may change the rate.
2. The authors only focus on the mini-max rate without considering the optimization performance. This is dangerous.
3. The result requires d to be large enough. How large? Would it be exponential with the sample size n?

For the empirical part:
1. More experiments (e.g., CIFAR-10, ImageNet) should be added.
2. The theory shows that the network size is not important; can the authors provide more experiments on this?
3. If there are experiences that do not perform 1/\sqrt{n} rate, could the authors provide some reasons?

Besides, I do not think adding too many derivations in the main body of the paper is good. This only harms readability. Why not provide some insights and defer all the details to the appendix?

In conclusion, while this paper touches on an important topic, it requires substantial improvements in terms of theoretical clarity, empirical evidence, and overall presentation.

**Questions:**

See limitations.

---

> ### Author Response · Authors · 2023-11-23
> **Overall response**
>
> We appreciate your insightful comments and thank you for your valuable suggestions for further improvement. Your points of view are very insightful.
>
> 1- We would like to note that, non-convexity and optimization difficulties of the problem are not considered in this work, and we assumed that global minimizers are available. But extensions are possible for future works using the ideas from these works (https://arxiv.org/abs/2205.04491 , https://ieeexplore.ieee.org/abstract/document/9875215)
>
>
> 2- Taking into account your valuable suggestions, we have introduced three additional network settings in the empirical part for the MNIST dataset. This addition aims to illustrate that network size is not a crucial factor in practice, aligning with our theoretical findings. Additionally, we conducted experiments with the CIFAR-10 dataset to augment our empirical results.
>
> 3- We are appreciative of you for raising this meticulous point regarding how large d would be. Now, we first have added $d>=10$ (based on the proof of Lemma~6 in Appendix) have added the following explanations after Lemma~6:
>
> By quantifying the lower bound of the packing number, it provides valuable insights into the capacity and potential complexity of these networks. For small values of $\\delta$, a sufficiently wide network becomes necessary.  This observation is particularly interesting as it provides valuable insights into selecting an appropriate width for the network based on the input dimension. The larger the input dimension $d$, the wider the network should be.
>
> 4- We would like to note that the challenges arising from the ReLU activation function, particularly in the absence of the separation of the estimation method and the function space, and without considering any smoothness for the function space, become more pronounced in the calculation of step 1 in Lemma 6, specifically in Equation 7. However, for other activation functions, this equation can be computed much more easily with simple arithmetic.

---

### Official Review · Reviewer_NyqE · 2023-11-02

**Soundness:** 2 fair
**Presentation:** 3 good
**Contribution:** 2 fair
**Rating:** 5
**Confidence:** 3

**Summary:**

This paper studied how much data is required to get a well-generalized neural network. It explores the generalization error of neural networks and suggests that it scales at $1/\sqrt{n}$ in terms of the sample size $n$. In detail, the authors provide both the upper and lower bounds for a specific neural network architecture and data distribution.  The authors also conducted some experiments to support their claims.

**Strengths:**

- This paper offers theoretical insights into the generalization error of neural networks, providing both upper and lower bounds.

- This paper not only establishes a bound for deep-ReLU networks but also empirically validates its findings.

**Weaknesses:**

- This paper appears to overclaim its results. Titled "HOW MANY SAMPLES ARE NEEDED TO TRAIN A DEEPRELU NEURAL NETWORK?" it investigates a much narrower setting than suggested. Firstly, the paper overlooks the training process, assuming that the optimizer can derive the best learner from a given function class. Therefore, the term "train" in the title is misleading. Secondly, the scope is limited to feed-forward neural networks with square loss, neglecting widely used structures like CNNs and transformers. In light of this, I recommend that the authors either provide additional commentary within the paper or refine the title and contributions to reflect the actual scope of the study.

- The paper adopts a teacher-student network setting with Gaussian input $x$ and noise $u$. The impact of the scale of $x$ and noise $u$ on the results in Theorem $1$ remains unclear. Additionally, the l1 constraint $v_1$ mentioned in Theorem 1 is not reflected in the final bound in (5), and the reason for this omission is not clear.

- Various papers demonstrate that a faster rate $O(1/n)$ is achievable under specific settings. It would be beneficial if the authors could address these findings, particularly discussing the conditions under which this faster rate could be realized for DNNs.

[1] https://dl.acm.org/doi/pdf/10.5555/3455716.3455772

[2] https://dl.acm.org/doi/pdf/10.5555/3455716.3455772

[3] https://proceedings.mlr.press/v54/mehta17a/mehta17a.pdf

[4] https://proceedings.neurips.cc/paper_files/paper/2015/file/acf4b89d3d503d8252c9c4ba75ddbf6d-Paper.pdf

**Questions:**

- What will happen if we consider a noiseless teacher-student model, i.e., $\sigma = 0$?

- This paper considers the setting where the input $x$ is isotropic. Will the results still hold when the covariance of the input $x$ is not identity?

- Each optimization algorithm has its own implicit bias. Will the results improve or worsen, given another specific data model and a specific algorithm?

---

> ### Author Response · Authors · 2023-11-23
> **Overall response**
>
> We appreciate your insightful comments and thank you for your valuable suggestions for further improvement.
>
> 1- Thank you for raising this point regarding the scope of our work. You are absolutely right. Now, we’ve changed the title of the paper to “How many samples are needed to train a ReLU feed-forward neural network? ” to make our setting more clear. For the future work, we are going to consider the setting for CNN and RNN as well.
>
> 2- Taking your thoughtful recommendation into account regarding directly reflecting the influence of $v_1$, we have revisited the final bound in Equation 5 to explicitly highlight the role of $v_1$. This revision not only enhances the clarity of our presentation but also provides a more transparent understanding of how the bound is influenced by the magnitude of $v_1$.
>
> 3- We would like to note that, non-convexity and optimization difficulties of the problem are not considered in this work, and we assumed that global minimizers are available. But extensions are possible for future works using the ideas from these works (https://arxiv.org/abs/2205.04491 , https://ieeexplore.ieee.org/abstract/document/9875215)
>
>
> 4- One fundamental part of our theoretical framework involves calculating the KL divergence between two normal distributions and subsequently establishing an upper bound for mutual information. As expressed in the proof of Lemma 4, the symbol $\\sigma$ denotes the standard deviation of these normal distributions. The derived upper bound for the mutual information (Lemma 4) is crucial for applying Fano’s method. Although our method doesn't directly apply to a noiseless teacher-student model, exploring the noiseless teacher-student model in itself would be intriguing.
>
> 5- You are absolutely right; the input $x$ is isotropic, and the covariance is identity. We would like to note that the assumption on $x$ is made for the sake of simplicity , in particular, in the calculation of Equation 7. However, extensions to other distributions are also feasible with more sophisticated methods. This assumption simplifies the calculation of Lemmas 11, 12, and 13.

---

> ### Comment · Reviewer_NyqE · 2023-12-05
>
> After reviewing the authors' responses, I acknowledge that they have addressed several of my concerns, though some issues remain unresolved, such as noise-less settings, in relation to previous work. In my opinion, the papers' quality lies between 3 to 5 currently. Consequently, I increased my score to 5 to reflect that.

---

### Official Review · Reviewer_NcCL · 2023-11-06

**Soundness:** 3 good
**Presentation:** 2 fair
**Contribution:** 3 good
**Rating:** 6
**Confidence:** 3

**Summary:**

The paper introduces an algorithm-free lower bound on the generalization error for deep ReLU neural networks by building up on Fano's inequality from information theory. The lower bound scales as $\frac{1}{\sqrt{n}}$ instead of the usual $\frac{1}{n}$ which proposes that when considering data-agnostic scaling laws, $-\frac{1}{2}$ exponent is more suitable than $-1$.

**Strengths:**

I like the step-by-step explanations of the paper and I like that it is focused on the rate $\sqrt{\frac{\log(d)}{n}}$ of the generalization error.
I was not familiar with Fano's inequality but the refs in the paper to Wainwright are to the point and it was sufficient to understand the results of this paper.

- Elegant use of Fano's inequality for ReLY nets
- It is interesting that deep networks follow the same rate as shallow networks. This is consistent with the literature.
- Simulations show better fit to $c_1 \frac{1}{\sqrt{n}} + \alpha $.

**Weaknesses:**

Although the authors do a good job of explaining the important components of their main result, the organization of the paper can be improved much further. See below for some points

- I'd prefer deferring the proof of theorem 1 to the appendix as it is not very insightful (In fact the proof is a straightforward combination of the lemmas).
- For the proof of Lemma4, the authors give an 8-step explanation which is simply reading through each step in the following inequalities. I'd prefer it if the most important step was emphasized and the other steps were left to the reader. For me the most important step there is the explicit integral formulation of $D_\text{KL}$ but I have no idea where $\frac{1}{\sigma^2}$ comes from.

On the other hand, I find the empirical evaluation limited as it is. There is only one architecture used for Figure 1 (a certain width and depth, which would be good to include in the caption). Is this proposed scaling law also valid for really narrow networks like say width 20? The rate here does not depend on the architecture, is it also reflected in practice?

Minor feedback:
- Eq (2), please specify that all activation functions are ReLU already here.
- In conclusion, please specify "networks" as deep ReLU feedforward networks.

**Questions:**

I am confused about the input data distribution. It is specified as $\mathcal{N}(0, I)$ at the beginning of Section 2 however layer then the authors introduce $L_2(\mathbb{P}_x)$ norm. Is $\mathbb{P}_x$ here Gaussian. If so, $h(x)$ comes from the metric $\rho$, not from the density of the distribution $\mathbb{P}_x$?

I would increase my score if the authors provided more experiments (see weaknesses) and/or provided an alternative in-depth discussion/intuition instead of the proof of 4.1.

---

> ### Author Response · Authors · 2023-11-23
> **Overall response**
>
> We appreciate your insightful comments and thank you for your valuable suggestions for further improvement  and we appreciate all of your encouragement and constructive feedback. We agree with all the suggestions and have carefully revised the paper, incorporating them.
> Highlights of the new version based on your comments:
>
> 1- We have introduced three different network settings in the empirical part, featuring three very narrow shallow-ReLU feed-forward neural networks. And demonstrate that the scaling is valid also for shallow- ReLU feed-forward neural networks.
>
> 2- Additionally, we conducted experiments with the CIFAR-10 dataset to provide more empirical results.
>
> 3- Addressing the suggestion to move the proof of Theorem~1 to the Appendix section was indeed a good point, and we have implemented this change.
>
> 4- Specifying the ReLU activation function at the beginning of defining our neural network model was also insightful, and we have followed your instructions in this regard.
>
> 5- Given that our work focuses solely on ReLU feed-forward neural networks, we have explicitly considered this point throughout the paper to enhance clarity.
>
> 6- To enhance clarity and readability, we have included the structure of the networks (specifying the number of hidden layers and the number of hidden nodes) in the captions.
>
> In the following, we address your questions in more detail.
>
>
> We appreciate your meticulous point of view regarding your question about the distribution of $x$.
> As mentioned in Section~ Problem formulation and main result, $\\mathbb{P}_{x}=\\mathcal{N}(\\boldsymbol{0},\\Identity)$, and we need this assumption in the calculation of Lemma~12. Regarding the relationship between $h(x)$ and $\\rho$:
>
> $h(x)$ is not directly coming from $\\rho$; instead $h(x)$ is used in the calculation of the $L_{2}(\\mathbb{P}_{x})-$norm.
> The density function $h(x)$, plays a role in quantifying the "distance" of two functions $\\widehat{f}, f^{*}$ with respect to the Gaussian distribution $\\mathbb{P}_{x}$. We've also added the term ``fixed'' to emphasize that the marginal distribution $\\mathbb{P}_{x}$ is fixed.
>
>  In response to your inquiry about the rationale behind $1/\\sigma^{2}$: In Lemma~9, we compute the Kullback-Leibler divergence between two distinct multivariate normal distributions. The term $1/\\sigma^{2}$ indeed arises from the definition of the KL divergence, considering that  $\\mathbb{P}{f{\\theta^{j}}}$ and $\\mathbb{P}{f{\\theta^{k}}}$ correspond to the distributions associated with ${f_{\\theta^{j}}}(x)$ and ${f_{\\theta^{k}}}(x)$, respectively. Reviewing Equation~6 could provide additional insight into this matter. The $\\sigma$ is the variance of $n$ i$\\cdot$i$\\cdot$d$\\cdot$\@ noises $u_{i}$. And based on the view of Lemma~9, we need to have a positive variance.

---

### Meta-Review · Area_Chair_2pCD · 2023-12-15

**Metareview:**

This paper attempts to analyze the sample complexity of deep neural networks with ReLU activations. While the effort is commendable, the claims are overstated given the assumptions made. Specifically, the theoretical lower bound on sample complexity relies on strong assumptions that the ground truth function belongs to a restricted function class representable by sparse neural networks. Additionally, while providing initial empirical analysis, more comprehensive experiments on larger benchmark datasets are needed to substantiate the conclusions. The lack of optimization analysis also limits the applicability of the results.

In summary, this paper shows some merits towards analyzing deep ReLU networks through a statistical lens. However, significant gaps remain both theoretically and empirically before the conclusions can be substantiated. I would encourage the authors to consider a major revision, and the paper could be much stronger if these concerns could be properly addressed.

**Justification For Why Not Higher Score:**

NA

**Justification For Why Not Lower Score:**

NA

---

### Decision · Program_Chairs · 2024-01-16

Reject